# Extending MESMER-X: A spatially resolved Earth system model emulator for fire weather and soil moisture

Yann Quilcaille[1], Lukas Gudmundsson[1], Sonia I. Seneviratne[1]

[1]Institute for Atmospheric and Climate Science, Department of Environmental Systems Science, ETH Zurich, Zurich, Switzerland.

*Correspondence to*: Yann Quilcaille (yann.quilcaille@env.ethz.ch)

**Abstract.** Climate emulators are models calibrated on Earth System Models (ESMs) to replicate their behaviour. Thanks to their low computational cost, these tools are becoming increasingly important to accelerate the exploration of emission scenarios and the coupling of climate information to other models. However, the emulation of regional climate extremes and water cycle variables has remained challenging. The MESMER emulator was recently expanded to represent regional temperature extremes in the new "MESMER-X" version, which is targeted at impact-related variables, including extremes. This paper presents a further expansion of MESMER-X to represent indices related to fire weather and soil moisture. Given a trajectory of global mean temperature, the extended emulator generates spatially-resolved realisations for the seasonal average of the Canadian Fire Weather Index (FWI), the number of days with extreme fire weather, the annual average of the soil moisture and the annual minimum of the monthly average soil moisture. For each ESM, the emulations mimic the statistical distributions and the spatial patterns of these indicators. For each of the four variables considered, we evaluate the performances of the emulations by calculating how much do their quantiles deviate from those of the ESMs. We argue that this framework can be expanded to further variables, given how it performs over a large range of annual indicators. Overall, the now expanded MESMER-X emulator can emulate several climate variables, including climate extremes and soil moisture availability, and is a useful tool for the exploration of regional climate changes and their impacts.

# 1 Introduction

Changes in climate extremes and water cycle variables have received an increased attention in recent years, for instance with dedicated chapters in the recent 6[th] Assessment Report of the Intergovernmental Panel on Climate Change (IPCC) (Seneviratne et al., 2021; Douville et al., 2021; Caretta et al., 2022). These assessments, also confirming the IPCC Special Report on 1.5°C of global warming (IPCC, 2018; Hoegh-Guldberg et al., 2018) showed that both climate extremes and changes in water cycle are substantially changing with increasing global warming, even when shifting from 1.5°C to 2°C of global warming. Evaluating the societal and economic impacts of these climate change requires different approaches (IPCC, 2014). They show that climate extremes and changes in water cycle affect many aspects of our societies, such as agriculture (Wiebe et al., 2015; Vogel et al., 2019; Hasegawa et al., 2021), the energy sector (Schaeffer et al., 2012; Perera et al., 2020), and human health (Libonati et al., 2022).

However, exploring regional changes in climate extremes and the water cycle, as well as their associated impacts, remains a challenging endeavour for multiple reasons. First, climate extremes occur with a lower frequency, thus robust analyses require larger samples to correctly represent their distributions (Kim et al., 2020). Besides, changes in the water cycle are more challenging to represent than changes in temperature (Allan et al., 2020). However, impacts of changes in climate extremes and water cycle conditions are essential to assess in the context of climate change projections, since they may also be of relevance to the emissions scenarios derived by Integrated Assessment Models (IAMs) (Stehfest et al., 2014). For instance, IAMs simulate the mitigation of climate change by using bio-energies with carbon capture and storage (BECCS) and afforestation. Yet, these nature-based solutions would be impacted by droughts and fires (Fuss et al., 2014; Smith et al., 2016; Anderson and Peters, 2016). Thus, accurately replicating regional changes in climate extremes and water conditions simulated by Earth System Models (ESMs) at a lower computational cost would help in exploring mitigation potentials and new emissions scenarios.

The MESMER emulator has been developed with this purpose, first for regional mean variables (Beusch et al., 2020a; Beusch et al., 2022b), and more recently also extended to the MESMER-X version representing TXx, the annual maximum temperatures (Quilcaille et al., 2022). Given a trajectory of global mean surface temperature, MESMER-X evaluates TXx for every land grid point of the Earth, over an

arbitrary number of emulations, reproducing the natural variability and the local statistical distributions of TXx. Each one of these emulations account for the spatial and temporal correlations in TXx. MESMER-X was trained on each available ESM of the Climate Model Intercomparison Project Phase 6 (CMIP6) over 1850-2100 (Eyring et al., 2016; O'Neill et al., 2016).

So far, climate emulators have focused on the representation of global properties (Nicholls et al., 2020; Nicholls et al., 2021), often without natural variability. Comparatively, there are few spatially-resolved climate emulators, and even less with natural variability (Link et al., 2019; Beusch et al., 2020a; Nath et al., 2021; Liu et al., 2023). There are even less emulators for climate extremes, either without representing natural variability (Tebaldi et al., 2020) or for a single ESM (Watson-Parris et al., 2022). Alternatives to emulators are also envisaged (Tebaldi et al., 2022). Good performances for the emulation of TXx over all available ESMs were shown for MESMER-X (Quilcaille et al., 2022), and its method has the potential to be extended to other climate extremes.

Here, we present new extensions that build on the MESMER-X framework to emulate annual indicators of interest for fire weather and soil moisture (Abatzoglou et al., 2019; Cook et al., 2020). These specific variables were chosen because they offer a range in statistical properties to stress-test the capacity of the emulator in various situations. While we focus here on the emulation of annual average of the soil moisture and the annual minimum of the monthly average of the soil moisture, these variable are related to changes in drought occurrence (Seneviratne et al., 2021). Furthermore, fire weather and soil moisture are both relevant to assess the potential of nature-based solutions to mitigate climate change, such as BECCS and afforestation (Wang et al., 2014; von Buttlar et al., 2017; Vogel et al., 2019; Lüthi et al., 2021). These variables are thus of high relevance for the further extension of the MESMER-X emulator.

## 2 General method of MESMER-X

### 2.1 MESMER-X as extension of MESMER

The spatially resolved emulator MESMER provides realizations of local annual mean temperature given a scenario of Global Mean Surface Temperature ($\Delta T$) (Beusch et al., 2020a). These emulations results from a local average response to the global climate signal and from a local term for the natural variability.

The forced response relies on pattern scaling (Tebaldi and Arblaster, 2014; Herger et al., 2015; Alexeeff et al., 2018). The natural variability is a stochastic term deduced from a temporal auto-regressive process with spatially correlated innovations. The model can be calibrated using climate model output, e.g. from the CMIP6 collection (Eyring et al., 2016) using the historical simulations and the SSP scenarios up to 2100 (O'Neill et al., 2016). Note that each ESM is calibrated separately to reproduce their individual responses. MESMER has already been used for different applications. For example, it can integrate spatial observational constraints to improve the local temperature projections (Beusch et al., 2020b). Furthermore, MESMER has also been coupled to the simple climate model MAGICC (Meinshausen et al., 2011), allowing for an efficient calculation of the local response to emissions scenarios, including not only uncertainties in modelling but also natural variability (Beusch et al., 2022b). An application of this coupling is the evaluation of the contributions of emitters to regional warming (Beusch et al., 2022a). A first extension of MESMER was achieved, allowing the emulation of monthly local temperatures (Nath et al., 2021).

The MESMER-X emulator is an extension of MESMER, dedicated to the representation of impact-related variables, including climate extremes, has already been described and showcased for annual maximum temperature (Quilcaille et al., 2022).

## 2.2 The MESMER-X approach: emulating spatially resolved climate variability by sampling from conditional distributions

The method used in the MESMER-X emulator can be summarized in two steps. First, MESMER-X replaces the pattern scaling of MESMER using conditional distributions with a more flexible "distribution" scaling (Tebaldi and Arblaster, 2014). Then, the training of the spatio-temporal correlations is similar to MESMER, albeit performed not on the residuals of the pattern scaling, but by projecting the sample onto a standard normal distribution using a probability integral transform.

We represent the climate variable $X_{s,t}$ for grid points $s$ and at annual time steps $t$. Typically, $X_{s,t}$ is deduced from CMIP6 historical and SSP scenarios, covering 1850-2100 and the whole Earth. The first assumption is that this variable can be represented locally by a probability distribution $\mathcal{D}$. For instance, block-extrema (e.g. annual maximum of temperature, monthly minimum of soil moisture) may be

represented by a Generalized Extreme Value distribution (GEV) (Coles, 2001). Similarly, averages (e.g. annual mean temperature) may be represented by a normal distribution. The second assumption is that this distribution $\mathcal{D}$ depends on variables expressing changes in global climate. Explicitly, the $p$ parameters $\alpha_{s,t,p}$ of $\mathcal{D}$ at grid points $s$ are functions $f_{s,p}$ of a matrix of global variables $\boldsymbol{V_t}$. The columns of the matrix $\boldsymbol{V_t}$ contain covariants, explanatory variables such as global mean temperature anomalies, while the rows of $\boldsymbol{V_t}$ correspond to time steps. The functions $f_{s,p}$ may be linear, quadratic, sigmoid or other functions of the covariants $\boldsymbol{V_t}$. In equation (1), we summarize how the probability $P$ of $X_{s,t}$ follows a distribution $\mathcal{D}$ conditional on global climate through its parameters $\alpha_{s,t,p}$ as functions $f_{s,p}$ of changes in global climate $\boldsymbol{V_t}$. We call configuration $E$ the choice of a distribution $\mathcal{D}$ combined with the equations for $f_{s,p}$.

$$E: \begin{cases} P(X_{s,t}) = \mathcal{D}(X_{s,t}|\alpha_{s,t,p}) \\ \alpha_{s,t,p} = f_{s,p}(\boldsymbol{V_t}) \end{cases} \tag{1}$$

In the case where $\mathcal{D}$ is a normal distribution and $f_{s,p}$ is linear on the mean and constant on the standard deviation of the distribution, this approach is equivalent to (Beusch et al., 2020a). Similarly, if $\mathcal{D}$ is a GEV, equation (1) is equivalent to the formalism introduced in the article showcasing MESMER-X (Quilcaille et al., 2022).

Equation (1) offers a large flexibility in terms of modeling. Using variables such as global mean surface temperature, radiative forcing or ocean heat content facilitates the representation of the most relevant processes within the Earth system. Using lagged variables such as the global mean temperature at $\Delta T_{t-n}$ or accumulated warming over the past $n$ years would also help in representing more advanced dynamics such as inertias in the water cycle. Such a capacity is of particular interest for overshoot scenarios. Yet, equation (1) has also its limits: it would not account for changes in local climate drivers (e.g. land-use, combination of individual radiative forcings) that would compensate at a global scale. Such effects may still be modeled (Nath et al., 2022), but are not integrated in this framework.

Nevertheless, these conditional distributions in each grid-cell are not enough, because they do not account for the spatio-temporal correlations. For instance, if the annual average soil moisture in one grid point happens to be lower than expected, the values in the adjacent grid points are probably also lower. To integrate these effects, we follow the approach of (Beusch et al., 2020a), that parametrizes internal climate

variability using the spatially autoregressive (SAR) noise model described in (Cressie and Wikle, 2011; Humphrey and Gudmundsson, 2019). The SAR model reproduces the temporal and spatial autocorrelation structure of the training data, using two components. Temporal correlations are represented by an auto-regressive process (equation 3). Spatial correlations are reproduced with spatially correlated innovations, randomly generated from a multivariate Gaussian with zero mean and covariance matrix derived from the training sample (equations 4 to 6). However, it assumes that the residual variability of equation (1) is stationary in time and is normally distributed. This is valid only if $\mathcal{D}$ is assumed to be a normal distribution and if it matches the considered sample. Here, we exploit that equation (1) provides the local distributions of the full sample. It means that we can use a probability integral transform to project the training sample $X_{s,t}$ on a standard normal distribution (Angus, 1994; Gneiting et al., 2007; Gudmundsson et al., 2012). We define $\mathcal{F}_{\mathcal{D}}$ as the cumulative distribution function (CDF) and $\mathcal{F}_{\mathcal{D}}^{-1}$ as the quantile function of $\mathcal{D}$ (or inverse CDF). We also write $\mathcal{N}$ the standard normal distribution, with 0 mean and unit variance. We write $\mathcal{F}_{\mathcal{N}}$ and $\mathcal{F}_{\mathcal{N}}^{-1}$ respectively as its CDF and inverse CDF. We then employ the probability integral transform, obtaining a normalized variable $\Phi_{s,t}$, where $\Phi_{s,t}$ has no trend and follows a standard normal distribution such that

$$\Phi_{s,t} = \mathcal{F}_{\mathcal{N}}^{-1}\left(\mathcal{F}_{\mathcal{D}}\left(X_{s,t}|\boldsymbol{V_t}, f_{s,p}\right)\right). \tag{2}$$

Note that equation (2) works equally well if $\mathcal{D}$ is a discrete distribution, as illustrated in Appendix 6.1. The normalized variable $\Phi_{s,t}$ are then characterized using an autoregressive process with spatially correlated innovations (Beusch et al., 2020a). In each grid point, a temporal auto-regressive process of first order is fitted on $\Phi_{s,t}$, with parameters $\gamma_{s,0}$ and $\gamma_{s,1}$, such that

$$\Phi_{s,t} = \gamma_{s,0} + \gamma_{s,1}\Phi_{s,t-1} + v_{s,t} \ with \ v_{s,t} \sim \mathcal{N}\left(0, \Sigma_v(r)\right). \tag{3}$$

The residuals $v_{s,t}$ represents spatially correlated innovations, drawn from a multivariate normal distribution with means 0 and covariance matrix $\Sigma_v(r)$ (Cressie and Wikle, 2011; Humphrey and Gudmundsson, 2019). Here, $r$ designs the ratio of geographical distance between points and a localization radius, and the next paragraphs explaining how $\Sigma_v(r)$ is obtained from the empirical covariance matrix.

The representation of interannual variability is discussed in Appendix 6.2. Using a first order auto-regression allows to analytically derive the covariance matrix $\Sigma_v(r)$ from the covariance matrix of the residual variability $\Sigma_\eta(r)$ (Cressie and Wikle, 2011), such that

$$\Sigma_v(r)_{i,j} = \sqrt{1 - \gamma_{i,1}^2} \cdot \sqrt{1 - \gamma_{j,1}^2} \cdot \Sigma_\eta(r)_{i,j} \tag{4}$$

where $i$ and $j$ are two grid points. In the simplest case, $\Sigma_\eta(r)$ would be the empirical covariance matrix $\tilde{\Sigma}_\eta$, estimated from $v_{s,t}$. However, in the usual settings of climate model emulation, the resulting covariance matrix is rank deficient since the number of spatial locations by far exceeds the number of considered time steps. To compensate for this rank deficiency, the empirical covariance matrix $\tilde{\Sigma}_\eta$ is regularized using localization, an approach well established in data assimilation (Carrassi et al., 2018).

The principle is to apply a function that conserves correlations for points relatively close to each other, but that shrinks distant points to zero. This localization is described in equation (5), with $\circ$ the Hadamard product and $G$ the Gaspari-Cohn function (Gaspari and Cohn, 1999) such that

$$\Sigma_\eta(r) = \tilde{\Sigma}_\eta \circ G(r) \tag{5}$$

Where the Gaspari-Cohn function, that takes $r$ as input, the ratio of the geographical distance between

two grid points and a localization radius $L$, is defined as

$$G(r) = \begin{cases} 1 - \frac{5}{3}r^2 + \frac{5}{8}r^3 + \frac{1}{2}r^4 - \frac{1}{4}r^5 & if\ 0 \le r < 1 \\ 4 - 5r + \frac{5}{3}r^2 + \frac{5}{8}r^3 - \frac{1}{2}r^4 + \frac{1}{12}r^5 & if\ 1 \le r < 2 \\ 0 & if\ 2 < r \end{cases} \quad with\ r = \frac{d}{L} \tag{6}$$

Equations (1-6) correspond to the full training of MESMER-X, with equation (1) to train the grid-cell specific conditional distributions, equation (2) as interface to the training of the spatio-temporal structure and equations (3-6) for this final part of the training. The emulations of climate extremes for a scenario,

typically over 1850-2100, require time series of anomalies in global climate $V_t$ over the period of the scenario, so that equation (1) generates the distributions at each grid point and each time step. Equation (3) generates an arbitrary number $n$ of realizations $\tilde{\Phi}_{s,t,n}$. The emulations $\tilde{X}_{s,t,n}$ are then the consequence of a back probability integral transform, as described in equation (7).

$$\tilde{X}_{s,t,n} = \mathcal{F}_\mathcal{D}^{-1}\left(\mathcal{F}_\mathcal{N}\left(\tilde{\Phi}_{s,t,n}\right)|V_t, f_{s,p}\right) \tag{7}$$

## 2.3 Configuration of MESMER-X

The performance of the emulator relies principally on the two assumptions made for equation (1): the choice of a distribution and the equations for its parameters, i.e. the configuration $E$. To assess and compare the performances, we use the ensemble Continuous Rank Probability Score ($CRPS$), a generalization of mean absolute errors for probabilistic forecasts. The $CRPS$ measures differences in the cumulative distribution functions of the emulations $\tilde{X}_{s,t,n}$ and of the training data $X_{s,t}$ (Hersbach, 2000; Wilks, 2011). It is also used to define the Continuous Rank Probability Skill Score ($CRPSS$) by comparing the $CRPS$ of a configuration $E$ to the $CRPS$ of a benchmark $E_0$. Both scores are commonly used in atmospheric sciences (Wilks, 2011; Jolliffe and Stephenson, 2012). Equations (8) and (9) respectively detail the calculation of the $CRPS$ and of the $CRPSS$, where $\mathbb{1}$ is the Heavyside step function.

$$CRPS^E\left(\tilde{X}_{s,t,n}, X_{s,t}\right)_{s,t} = \int_{-\infty}^{+\infty} \frac{1}{n} \sum_n \left[\mathbb{1}\left(X \geq \tilde{X}_{s,t,n}\right) - \mathbb{1}\left(X \geq X_{s,t}\right)\right]^2 dX \tag{8}$$

$$CRPSS_{s,t}^E = 1 - CRPS_{s,t}^E \Big/ CRPS_{s,t}^{E_0} \tag{9}$$

Here we consider a fit with a stationary distribution as the benchmark. A high $CRPS$ for this benchmark means that the differences between the cumulative distribution functions are too big, which implies that a stationary distribution does not correctly reproduce the statistical properties of the training sample, while a distribution reproducing perfectly the training sample would have a $CRPS$ of zero (Hersbach, 2000), as illustrated with Figure A. 1, in the Appendix 6.3. A high $CRPSS$ for a proposed configuration means that it improves the reproduction of the statistical properties of the sample. To simplify the comparisons, the $CRPSS$ is averaged over space, time and scenarios.

## 3 Emulations for fire weather

Many factors contribute to the burned area by wildland fires. Agricultural expansion and landscape fragmentation tend to decrease the burned area (Andela et al., 2017), though the global wildfire danger itself tends to increase (Jolly et al., 2015). The strong wildfires observed over the past years had their risk of happening increased by climate change (Li et al., 2019; van Oldenborgh et al., 2021), because it affects

the conditions to have ignition and spreading of wildfires. Such conditions are termed as fire weather. The strengthening of the fire weather favours longer-lasting and more intense fires (Abatzoglou et al., 2019; Ranasinghe et al., 2021; Seneviratne et al., 2021). The effect of climate change on fire weather is especially strong for the extreme events of fire emissions and burned area (Jones et al., 2022; Ribeiro et al., 2022). The Canadian Fire Weather Index (FWI) is one of the indices used to evaluate how daily temperatures, precipitations, wind and relative humidity are locally conducive to the occurrence and spread of fires (Van Wagner, 1987; Abatzoglou et al., 2019). The FWI is relevant to investigate the impacts of fire weather, thanks to its relationships to the burned area (Bedia et al., 2015; Abatzoglou et al., 2018; Grillakis et al., 2022; Jones et al., 2022).

In the following we adapt the MESMER-X framework presented in Section 2.2 for annual indicators of the FWI. We describe the data used for the training and emulation of the fire weather (Section 3.1), then extend the method of MESMER-X to the emulation of seasonal average of the FWI (Section 3.2) and the number of days with extreme fire weather (Section 3.3).

## 3.1 Data for the annual indicators of the Fire Weather Index

Here we consider annual indicators of the FWI computed using CMIP6 data (Quilcaille et al., 2023). The algorithm used combines adjustments from various packages to the original algorithm (Van Wagner, 1987), each aiming at extending the applicability of the FWI (Quilcaille et al., 2023). The calculations were applied over the historical period and the Shared Socioeconomic Pathways scenarios used by ESMs (O'Neill et al., 2016). All runs with available daily temperature, relative humidity, wind speed and precipitations were computed, in order to maximize the number of ensemble members for the ESMs, reaching a total of 1486 runs. The daily FWI is regridded onto a common 2.5° x 2.5° longitude-latitude grid using second order conservative remapping (Jones, 1999; Brunner et al., 2020).

The data presented by (Quilcaille et al., 2023) are available in four annual indicators that represent different aspects of fire weather: the local annual maximum of the FWI ($FWIxx$), the number of days with extreme fire weather ($FWIxd$), the length of the fire season ($FWIls$) and the seasonal average of the FWI ($FWIsa$). Here we consider only $FWIxd$ and $FWIsa$, for a greater variety in our approaches and

less repetitions. $FWIxd$ is defined by counting the number of days exceeding each year a local threshold defined as the 95$^{th}$ percentile over 1850-1900, while $FWIsa$ is defined as the local annual maximum of a 90-day running average over time.

## 3.2 Emulation of the seasonal average of the Fire Weather Index

To emulate $FWIsa$, the first step is to propose an appropriate distribution as explained in Section 2. $FWIsa$ is defined as the annual maximum of a 30-days running average over time. As a block-maxima, a GEV distribution may represent correctly the distribution of $FWIsa$ (Coles, 2001). However, the 30-days running average may be a reason to use a normal distribution. The second step for emulations is to propose evolutions of the parameters of the distributions. From a physical perspective, $FWIsa$ is a product from daily time series of temperature, relative humidity, precipitations and wind speed, which may support relatively elaborated expressions. From a statistical perspective, the evolutions of $FWIsa$ with $\Delta T$ shows a relatively linear dependency of the average and sometimes on the spread of the samples. Some grid points show ground for quadratic dependencies, especially in South America. We represent in Figure 1 all the configurations investigated. For a normal distribution, the parameters $\alpha$ introduced in equation (1) are the location and scale, written respectively $\mu$ and $\sigma$ in Figure 1, corresponding to the mean and standard deviation of the distribution. For a GEV distribution, the parameters $\alpha$ are the location, scape and shape, written respectively $\mu$, $\sigma$ and $\xi$ in Figure 1.

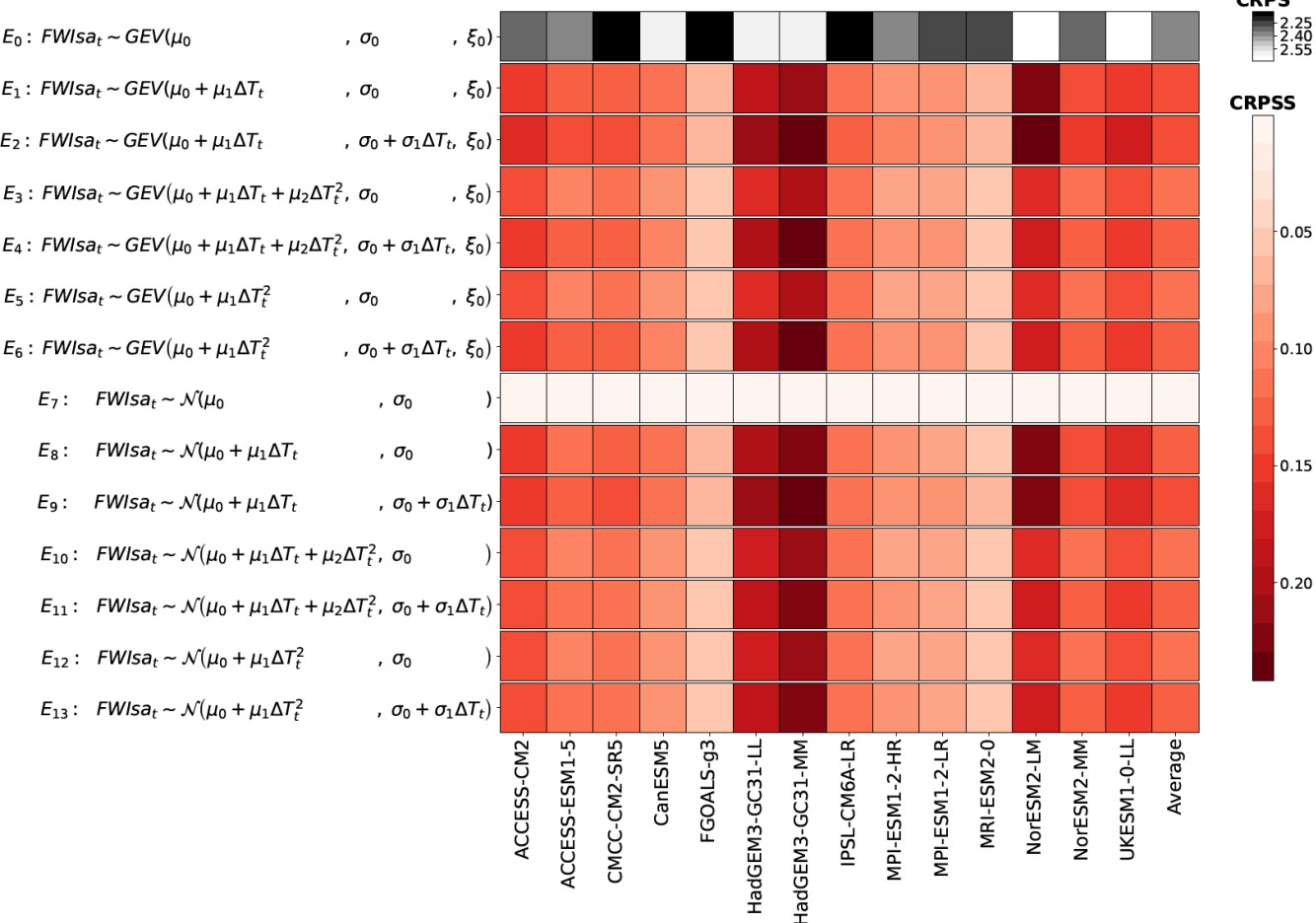

**Figure 1: Selection of the configuration for the seasonal average of the FWI ($FWIsa$).** For each ESM, the CRPS and CRPSS are averaged over space, time and scenarios. The darker is the colour of a cell, the better is the configuration at reproducing the distribution of the ESM. The upper row (white to black) corresponds to the CRPS of the configuration used as benchmark. A higher CRPS (lighter colour) indicates that the stationary distribution used as benchmark does not reproduce well the distribution of the ESM. The next rows (white to red) correspond to the CRPSS of the tested configurations, relatively to the benchmark. A higher CRPSS (darker colour) indicates that the proposed configuration improves the reproduction of the distribution of the ESM.

A stationary GEV distribution is used as benchmark for all the other configurations. Comparing this benchmark $E_0$ to a stationary normal distribution ($E_7$) show that the two of them are equivalent as benchmark. We note that ESMs with higher CRPS tend to have higher CRPSS. For these ESMs, stationary distributions are worse at representing their potentially stronger climate signal, meaning that the

improvement over a stationary distribution would be relatively higher. We note that the two configurations with the best average CRPSS are $E_2$ and $E_9$, that differ only by their distribution. Both have linear terms on the location and the scale. $E_2$ performs slightly better than $E_9$ because some points present skewed distributions, better represented by a GEV distribution. Using quadratic evolutions tend to increase the performance of the fit in only a minority of grid points while decreasing the performance over the rest of the land area. For this reason, the next results shown in Figure 2 and Figure 3 are performed using configuration $E_2$. We point out that the local performances for this configuration are shown in the Appendix 6.4, along with those of the other variables emulated.

$$E_2: FWIsa_{s,t} \sim GEV\left(\mu_{s,0} + \mu_{s,1}\Delta T_t, \sigma_{s,0} + \sigma_{s,1}\Delta T_t, \xi_{s,0}\right) \tag{10}$$

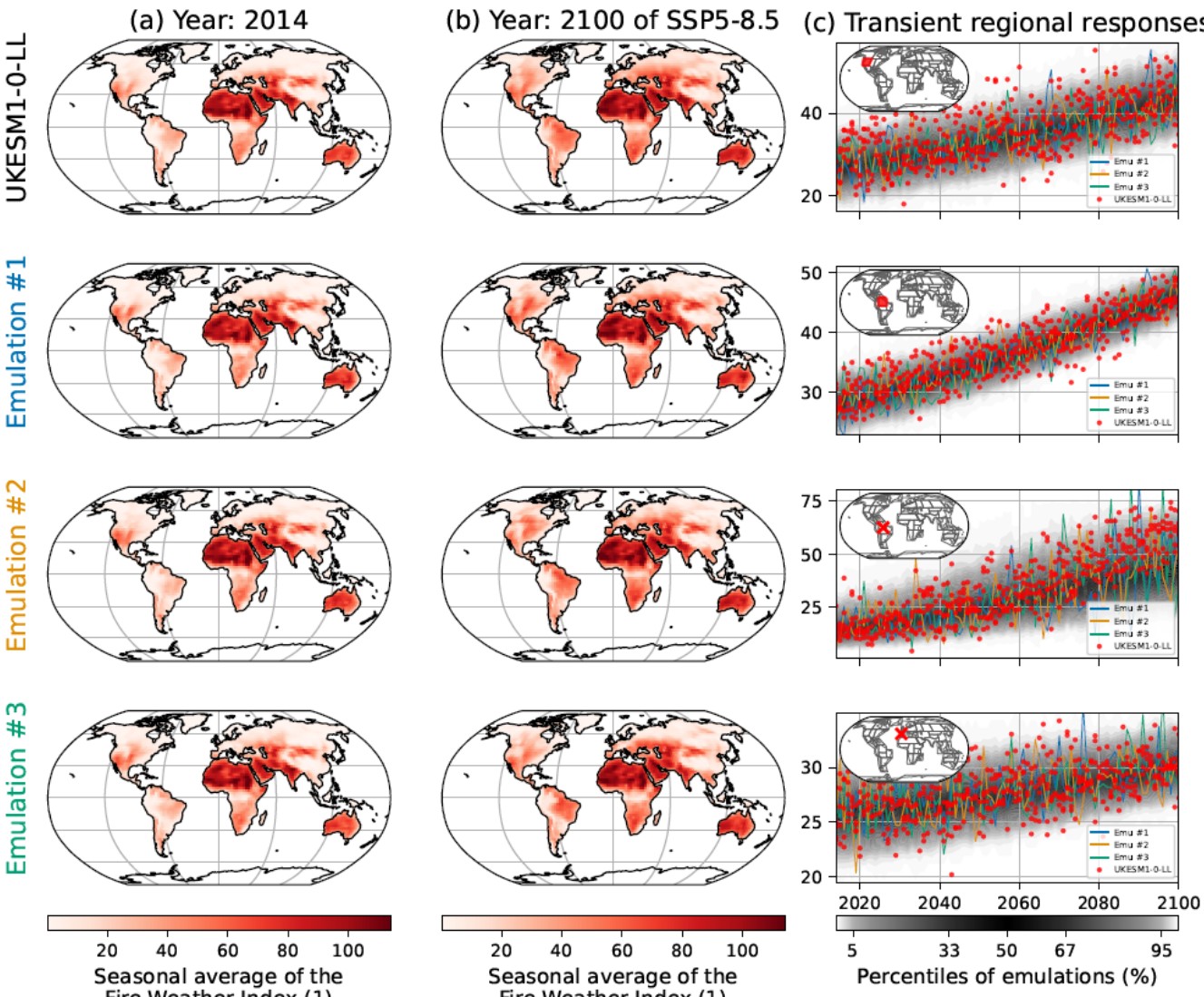

**Figure 2: Examples of results for the emulations of the seasonal average of the FWI ($FWIsa$) under UKESM1-0-LL.** The left column (a) represents maps of $FWIsa$ in 2014 according to UKESM1-0-LL on the first row, while the three following rows correspond to three emulations chosen randomly in the full set. The middle column (b) reproduces the same structure, although in 2100 of SSP5-8.5. The third column (c) shows time series of UKESM1-0-LL, the three emulations used for maps, but also the full spread of the emulations (shaded area). The rows correspond from top to bottom to the West of North America, the North of South America, a grid point in Amazonia close to Manaus and a grid point in Portugal close to Lisbon.

We show examples of emulations in Figure 2a,b, illustrating the capacity of the emulator, here on UKESM1-0-LL shown on the top row. Be it in 2014 or in 2100, the three random emulations on the three

other rows reproduce the spatial patterns of the ESM. There are some minor differences that are related to internal variability (ESM) and the stochastic representation thereof (emulator). Figure 2c illustrates the transient responses of $FWIsa$ of the emulations and of the ESM over the course of SSP5-8.5. Note that each row of column (c) is a chosen grid point or regional average. The red dots correspond to the realizations by UKESM1-0-LL for all ensemble members available, while the black shaded area represents the distribution of emulations. Over 2014-2100, the realizations by UKESM1-0-LL remain mostly within the range of the emulations, except for the third row that corresponds to a grid point close to Manaus in Amazonia. Figures similar to Figure 2 are provided in the Appendix 6.5 for low and mid warming scenarios.

Figure 3 provide more details on the deviation of quantiles of MESMER-X for each ESM and land region (Iturbide et al., 2020), thereafter called ESMs x regions. Overall, the panel (a) shows that the quantiles at 97.5% of the emulations is lower than those of the ESMs, but higher for the quantiles at 2.5%, shown in panel (c). This underdispersion is common for spatial emulators (Beusch et al., 2020a; Quilcaille et al., 2022), and regional aggregation contribute to this effect. For the quantile 97.5%, the deviation of quantiles range from +1.5% to -7.3%, with an average at -1.5%. In other words, the quantile 97.5% of the emulations woud actually rather be at 96% on average when compared to the ESMs. For the median, the deviations range from -8.4% to 13.3%, with an average of -0.3%. Finally, the deviations at the quantile 2.5% range from -1.2% to 16.0%, with an average at 2.2%. We note that the stronger deviations on the median occur when replicating NorESM2-LM. Because MESMER-X only aims at replicating the behaviour of ESMs, it cannot be used to diagnose the reasons for this difference. First analysis might suggest that the response of $FWIsa$ to $\Delta T$ is stronger than for other ESMs and that quadratic terms in the configurations may have a greater importance for this model.

In summary, the deviations of quantiles is less than 5% in absolute value for at least 92% of the ESMs x regions. Respectively for the quantiles 97.5%, 50% and 2.5%, these proportions of ESMs x regions below 5% of deviation are 98%, 93% and 92%.

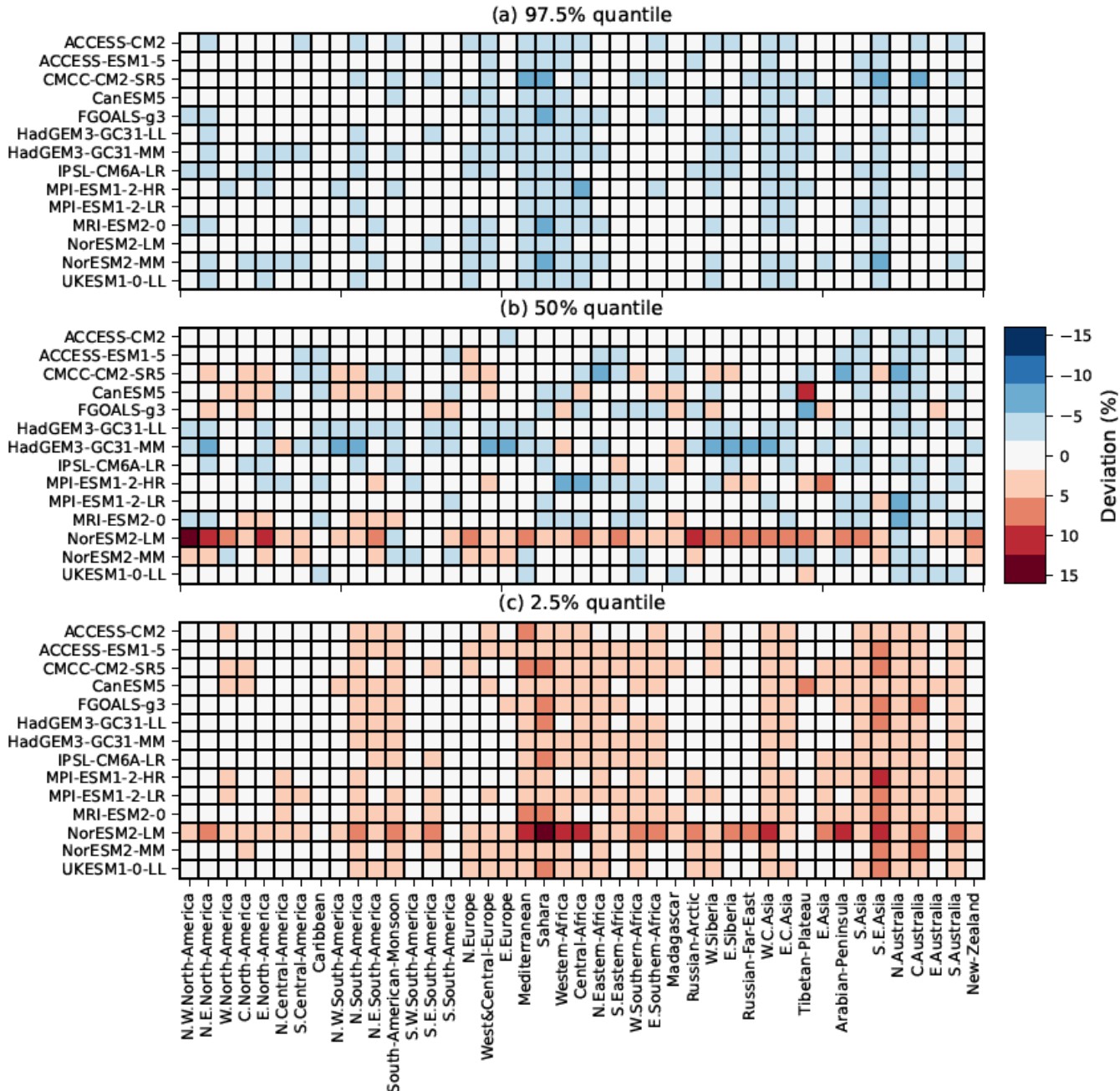

**Figure 3: Deviations of quantiles for the seasonal average of the FWI ($FWIsa$) at each ESM and each AR6 regions.** A positive deviation of quantiles (red) indicates that the quantile of emulations is higher than the one of the ESM, found by counting how often the ESM crosses the threshold set by the emulations. The deviation is calculated on all available scenarios. The upper panel (a) shows the deviations for the quantile 97.5%, the middle panel (b) for the median and the lower panel (c) for the 2.5% quantile.

### 3.3 Emulation of the number of days with extreme fire weather

For emulating the number of days with extreme fire weather ($FWIxd$) we consider the Poisson distribution, since it describes number of events occurring over a fixed period (Coles, 2001). Using this distribution implicitly assumes that the events are independent of each other, which is not exactly the case here. Assuming that a day matches the criteria for extreme fire weather (Quilcaille et al., 2023) for instance during the fire season, there are higher chances to have the next days also matching this criteria,

compared to a period out of the fire season. Nevertheless, we choose this distribution because of its relative simplicity. Similarly to $FWIsa$, linear and quadratic terms are investigated given the physical basis and the observed responses to $\Delta T$ (Jain et al., 2022). The comparison of the envisioned configurations are summarized in Figure 4. Here, the parameters $\alpha$ introduced in equation (1) are the rate $\lambda$ and a shift $\mu$. The training of the distribution gains in freedom using this shift of the distribution by $\mu$,

with its mean becoming $\mu + \lambda$, while the variance remains $\lambda$.

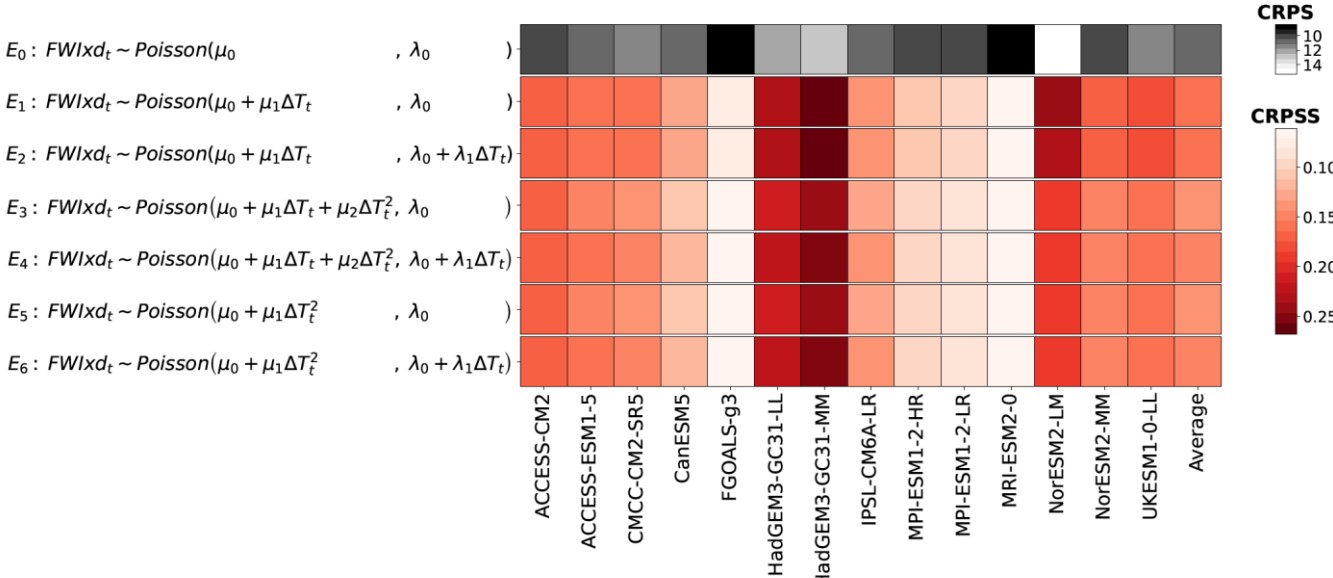

**Figure 4: Similar to Figure 1, although for the number of days with extreme fire weather ($FWIxd$).**

A stationary Poisson distribution is used as benchmark, showing a range of performances in CRPS greater for $FWIxd$ (9 to 15) greater than the one obtained for $FWIsa$ (2.1 to 2.6). Because the higher is a CRPS, the worse is the distribution at representing the training sample, two results can be deduced. First, stationary GEV distributions are much better at reproducing $FWIsa$ than stationary Poisson distributions are at reproducing $FWIxd$. It may be because $FWIxd$ has stronger responses to climate change than

$FWIsa$, meaning that stationary distributions, Poisson or GEV, cannot correctly reproduce these evolutions. It may also be because the shape of a Poisson distribution cannot reproduce the shape of the observed $FWIxd$ as well as a GEV can for $FWIsa$. From Figure 4, we observe that the best configuration is $E_1$, with only a linear evolution of the location of the distribution. The configuration $E_2$ had almost the same quality, although not as good for CMCC-CM2-SR5, MPI-ESM1-2-HR and NorESM2-LM. Like

$FWIsa$, few grid points, especially in South America would benefit from a quadratic term. Though, increasing the complexity of the functions for the parameters improved the fit only in few grid points, while decreasing the performances in many other places. The configuration $E_1$ has the best overall performances in spite of its simplicity, thus we use this one for the results presented in Figure 5 and Figure 6.

$$E_1: FWIxd_{s,t} \sim Poisson\left(\mu_{s,0} + \mu_{s,1}\Delta T_t, \lambda_{s,0}\right) \tag{11}$$

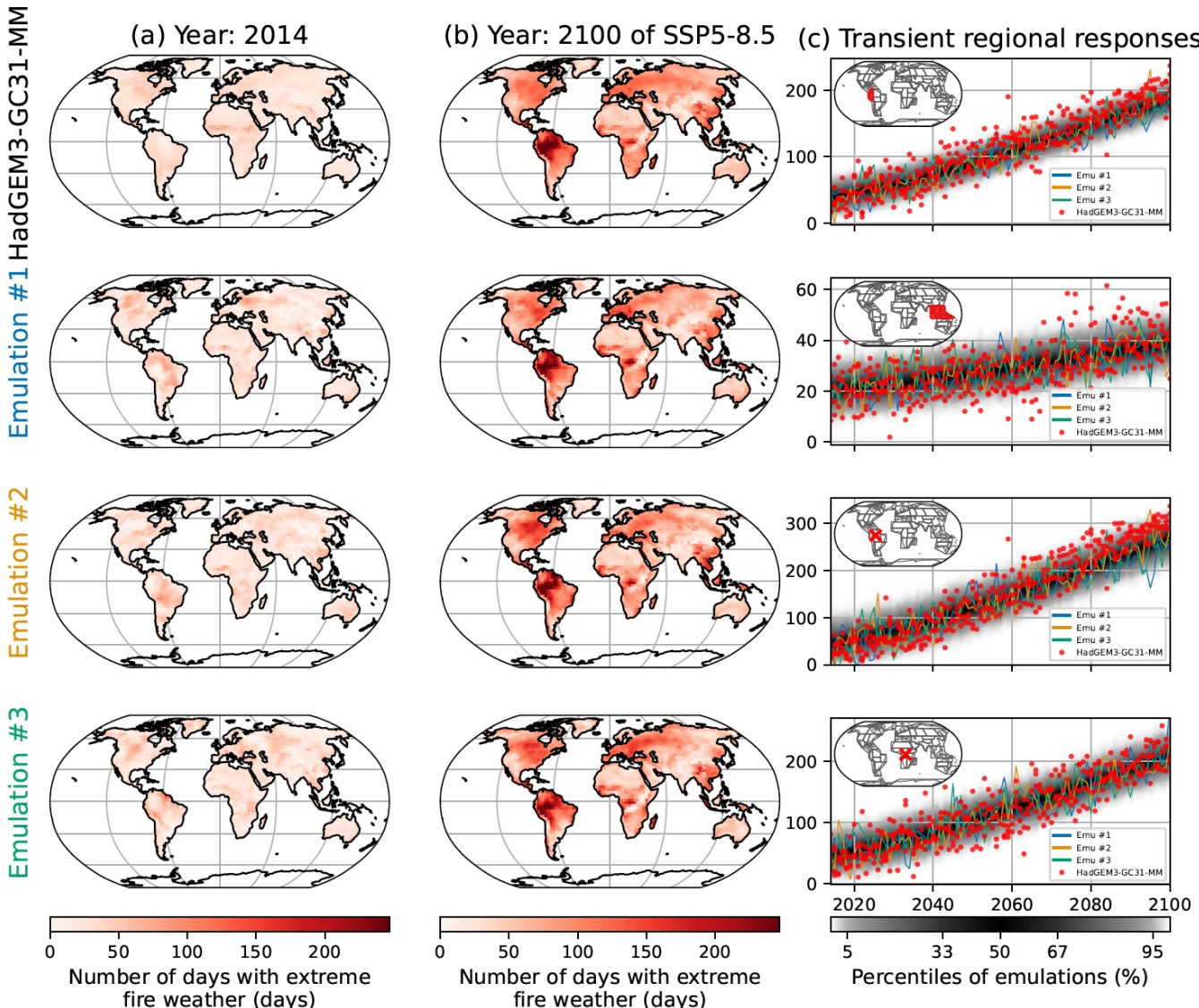

**Figure 5: Similar to Figure 2, although for the number of days with extreme fire weather ($FWIxd$) under HadGEM3-GC31-MM.** The rows correspond from top to bottom to the North-West of South America, South-East Asia, a grid point in Amazonia encompassing the Jaú National Parc and a grid point in Democratic Republic of Congo encompassing the Salonga National Park.

Just like Figure 2, we show in Figure 5 examples of outputs for the emulation of $FWIxd$. The spatial patterns are overall well respected, be it in 2014 or in 2100 (Figure 5a, b). There are indeed some differences due to natural variability. For instance, in 2014 (Figure 5a), HadGEM3-GC31-MM returns higher $FWIxd$ to the south of Sahel, but lower in South America. In 2100 (Figure 5b) in the centre of

Africa and in South-East Asia, we see differences in these patterns, though the emulations always relatively similar. Looking at the transient regional responses (Figure 5c), the two regions and the two grid points represented show that HadGEM3-GC31-MM and the emulations have similar evolutions, with the distribution of the emulations correctly encompassing the dispersion of the ESM. We point out one exception in these time series on the third row. This grid point in Amazonia shows that the $FWIxd$ of HadGEM3-GC31-MM increases faster than the emulations replicates. The same effect appears on the first row, although to a lesser extent. Some grid points in South America would benefit from a quadratic response to $\Delta T$, although Figure 4 shows that a linear response has better overall performances. Figures similar to Figure 5 are provided in the Appendix 6.6 for low and mid warming scenarios.

We show in Figure 6 the regional performances of the emulator by assessing the deviations of its quantiles to the ESM. On average, the emulators are -2.8% lower than ESMs for the 97.5% quantile, 4.4% higher for the median and 1.41% higher for the 2.5%. Overall, the emulators show lower performances in some regions such as South-East Asia, as shown in Figure 5, or to mimic some models such as NorESM2-LM. Reasons for the latter cannot be pinpointed to specific processes, as explained in Section 3.2. We observe that the median shows overall lower performances than for the tails of the distribution.

To summarize the performances on $FWIxd$, the deviations of quantiles are less than 5% in absolute value for 95% of the ESMs x regions at the 97.5% quantile. At the 2.5% quantile, the fraction of these ESMs x regions below 5% of deviation decreases to 92%. However, at the median, only 54% of the ESMs x regions are below 5% of deviation. A potential explanation may be the temporal dependence of the events, not respecting one of the conditions for the use of a Poisson distribution. As detailed at the beginning of this section, this work using a Poisson distribution is a first attempt with discrete distributions. Using other distributions that would not assume independent events may improve these results but would require a higher degree of complexity.

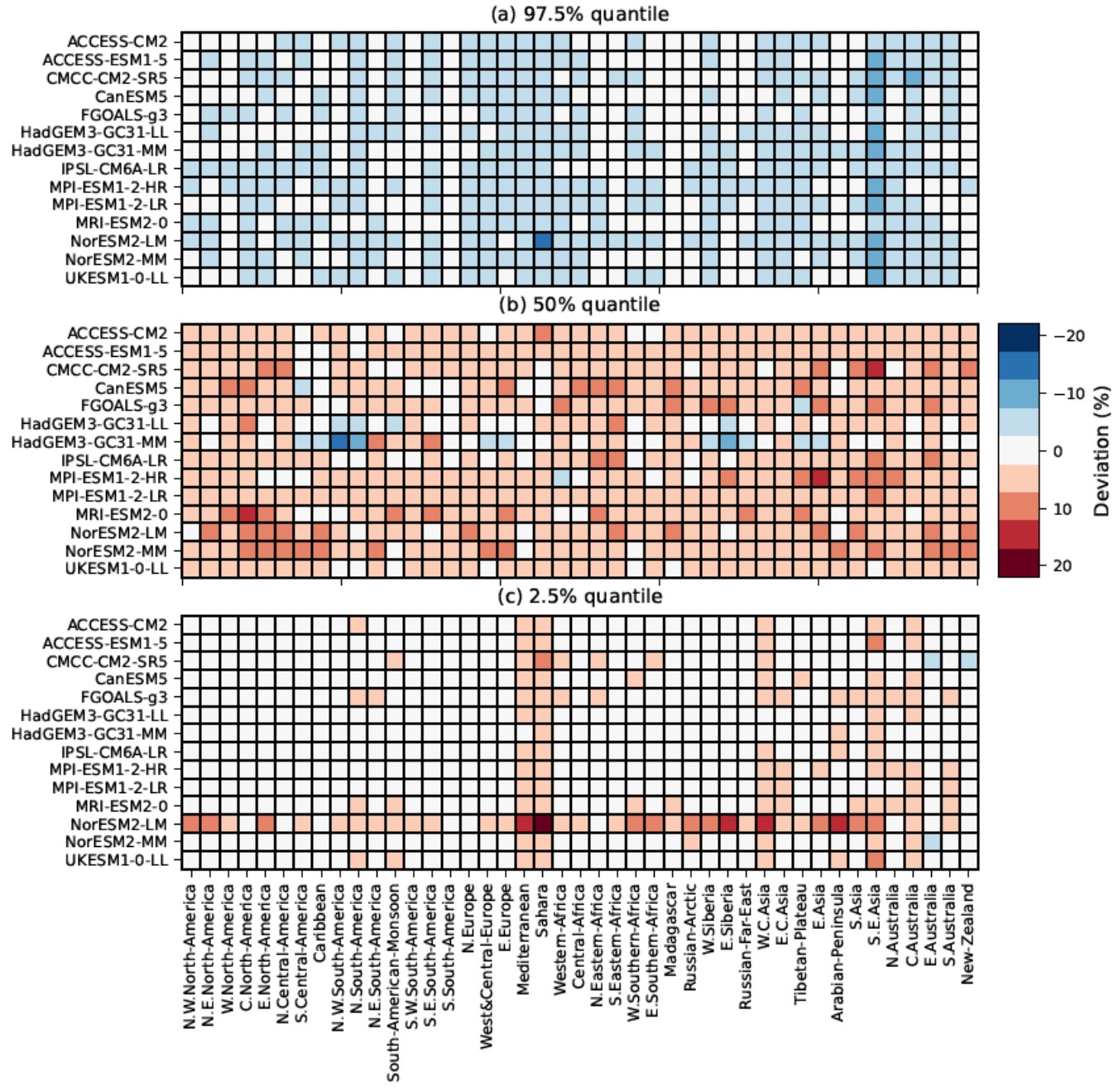

**Figure 6: Similar to Figure 3, although for the number of days with extreme fire weather ($FWIxd$).**

## 4 Emulations for soil moisture

### 4.1 Data for the annual indicators of soil moisture

We base the annual indicators for soil moisture on the total soil moisture content (CMIP6 variable *mrso*).
Ideally, soil moisture in the root zone would be more relevant to investigate droughts. Thus, soil moisture
in soil layer (CMIP6 variable *mrsos* or *mrsol*) would have been more adapted (Qiao et al., 2022).
Similarly, the total soil moisture content includes all water phases, thus frozen soil moisture as well. We
deem that the total soil moisture content remains relevant for droughts, in regions without high frozen
soil moisture, that is to say not higher latitudes or mountainous regions like the Himalaya. Nevertheless,
a majority of ESMs only provide the total soil moisture content, thus choosing this variable ensures that
the capacity of the emulator can be evaluated on more models and ensemble members.
Before computation of the annual indicators, the total soil moisture content of all available CMIP6 runs
is regridded onto a common 2.5° x 2.5° longitude-latitude grid using second order conservative remapping
(Jones, 1999; Brunner et al., 2020).
Two annual indicators are deduced from the total soil moisture content. By averaging this variable over
the year, we obtain the annual average of soil moisture (*SM*). Besides, we calculate the average over each
month and deduce their minimum, thus obtaining the annual minimum of the monthly average soil
moisture (*SMmm*). These two annual indicators are both relevant to assess the evolutions of droughts
(Cook et al., 2020). The annual average *SM* provides an indicator for the whole year, while the annual
minimum *SMmm* informs about the worst period of the year.

### 4.2 Emulation of the annual average of soil moisture

As for the fire weather, the first step for emulation is to choose a proper distribution. As an annual average,
*SM* may be represented by a normal distribution according to the central limit theorem. The second step
is to propose evolutions for the parameters. The impact of global temperature on the local total soil
moisture content is not as straightforward as for the two former cases. Many processes affect this variable,
through evapotranspiration, precipitations or runoff (Cook et al., 2020). Some regions show a decreasing
trend in the soil moisture, others an increase (van den Hurk et al., 2016; Qiao et al., 2022). A first choice

could be to propose a linear evolution on the mean (Greve et al., 2018). Though, going through local

responses of $SM$ to $\Delta T$ show that they may often be non-linear, e.g. following a sigmoid response. Such responses are characteristic of an evolution between two regimes, illustrated in Figure 7.

Another feature of these local responses are lagged effects. The response under SSP1-2.6 (blue points) decreases faster with $\Delta T$ than SSP2-4.5 (dark green points). The same effect happens with SSP3-7.0 (brown points) and SSP5-8.5 (orange points). The faster the warming increases and the slower is the slope

in the response of $SM$ to $\Delta T$. A potential explanation would be that different timescales are at play in the response of $SM$ to $\Delta T$. In high warming scenarios, the $\Delta T$ increases relatively fast to the response of $SM$ to the change in $\Delta T$, not letting the $SM$ stabilize. In SSP1-2.6 however, the $\Delta T$ stabilizes, allowing the $SM$ to stabilize as well. To a broader extent, this effect is related to the response of the whole water cycle, with rapid adjustments and slow feedback responses, both in precipitations and evapotranspiration(Allan

et al., 2020). Different methods may be used to represent the effect of different timescales, such as lagged variables or impulse response functions. Here, as a first attempt to reproduce this effect, we will test in the configuration a lagged variable using the $\Delta T$ at the former year. This lagged variable is obtained by shifting the $\Delta T$ of the ESM by one year. From a modeling perspective, having both $\Delta T_t$ and $\Delta T_{t-1}$ is equivalent to having the value at year $t$ and its first derivative.


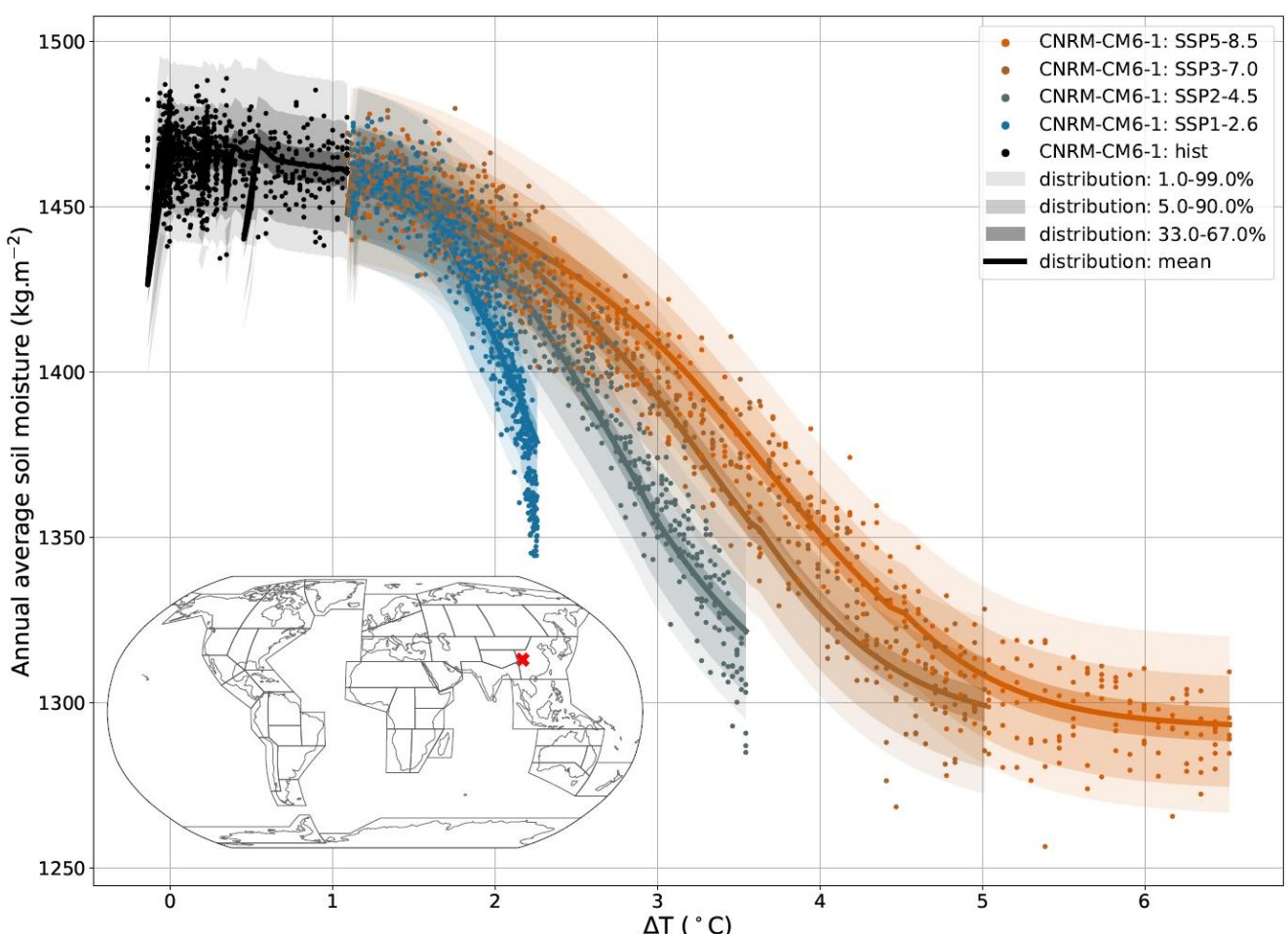

**Figure 7: Example of local response of the annual average soil moisture ($SM$) to ΔT under CNRM-CM6-1.** The grid point is in Sichuan, in the vicinity of Chengdu, the same one shown in Figure 9, column (c), fourth row. The distribution shown follows the configuration $E_4$ described in equation (12).

Figure 8 shows the results for all the tested configurations, with the coefficients $\mu$ and $\sigma$ corresponding respectively to the location and the scale of the normal distribution. For all ESMs except ACCESS-ESM1-5 and CNRM-ESM2-1, the best performances according to the CRPSS are met with $E_4$. For these two other ESMs, the better configuration $E_5$ differs only from the linear response on the standard deviation of the distribution. We note that introducing a logistic response on the mean ($E_3$) improves the performances

in a large majority of the grid points, more than a linear effect ($E_1$). Introducing the lagged effect has an effect not as clear ($E_4$), because the CRPSS is averaged over time and scenarios. Given these results, we

choose to use the configuration with the best performances for most ESMs. The results presented in Figure 9 and Figure 10 will then use the configuration $E_4$.

$$E_4: SM_{s,t} \sim \mathcal{N}\left(\mu_{s,0} + \mu_{s,L} + \frac{\mu_{s,R} - \mu_{s,L}}{1 + exp\left(\lambda_{s,1}\Delta T_t + \lambda_{s,2}\Delta T_{t-1} - \mu_{s,\varepsilon}\right)}, \sigma_{s,0}\right) \tag{12}$$


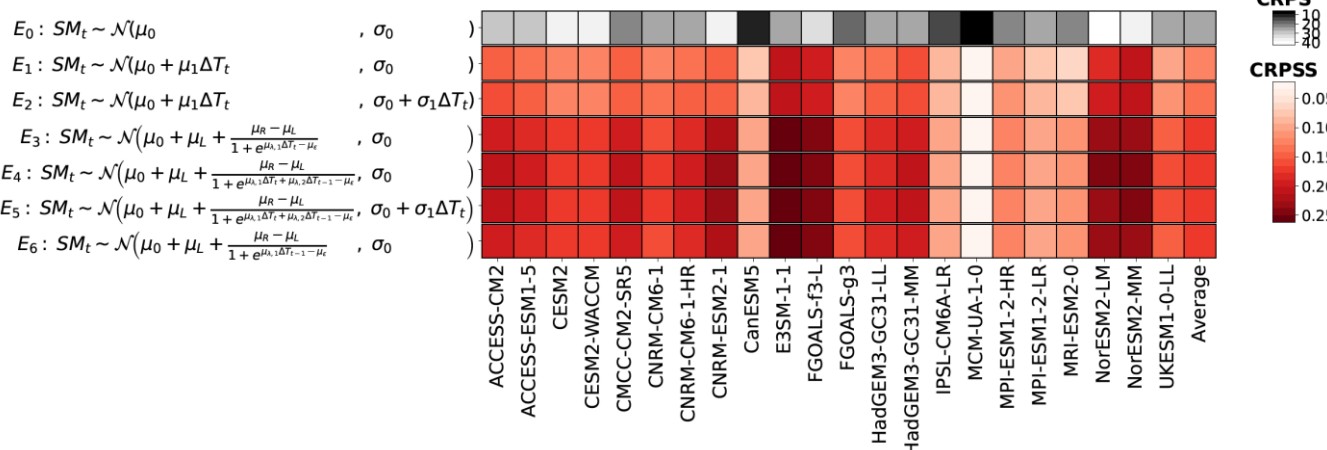

Figure 8: Similar to Figure 1, although for the annual average soil moisture ($SM$).

In Figure 9, we illustrate the emulations of $SM$ for CNRM-CM6-1. Just like for $FWIsa$ (Figure 2) and
$FWIxd$ (Figure 5), the spatial patterns are correctly reproduced. Note that the mean climate signal is dominating and thus effects of internal variability are hardly visible. The time series in Figure 9c show, however, that the natural variability is in general well reproduced over the course of SSP5-8.5. In the region West & Central Europe, the ESM seems to be often below the 5% quantile of the emulations, especially around 2050. In the region West of Southern Africa, the spread of the distribution is relatively
large, but represents relatively well the spread of the ESM in this region. We point out that the six ensemble members shown in this figure combined to the large regional spread show many points relatively far from the 90% range of the emulations, but the repartition of the realizations by CNRM-CM6-1 in this region is still well respected. Figure 9c shows however that some aspects of the dynamics are not entirely captured by the emulator, such as the short increase over 2040-2050 in Brazil. It may
indicate that choosing the $\Delta T$ over the former year is not good enough to represent lagged effects, or that

there are additional processes that cannot be represent as such by MESMER-X. Figures similar to Figure 9 are provided in the Appendix 6.7 for low and mid warming scenarios.

In Figure 10, we show the deviations on the regional quantiles of the emulations in each ESM x region. Just like with $FWIsa$ (Figure 3) and $FWIxd$ (Figure 6), the emulations are overall underdispersive. The 97.5% quantile (panel a) shows that the emulations have their quantiles -1.9% on average lower than their ESMs counterparts, up to -10.3%. There, the lower performances of MESMER-X occur in Sahara and in South-East Asia. Panel (b) shows that the median of emulations are on average 0.4% higher than the ESMs, these deviations ranging from 18.9% to -12.7%. We note lower performances in regions of Australia and in the Caribbean. Finally, the deviations on the 2.5% quantile shows that the emulations are on average 1.5% higher than the ESMs, up to 15.7% of deviations. The emulator for FGOALS-g3 exhibits lower performances than for other ESMs, although the reason for this remains unclear.

As a summary on the performances of the emulations of $SM$, the deviations are limited to 5% in 96% of the ESMs x regions at the 97.5% quantile, 88% at the median and 97% at the 2.5% quantile.




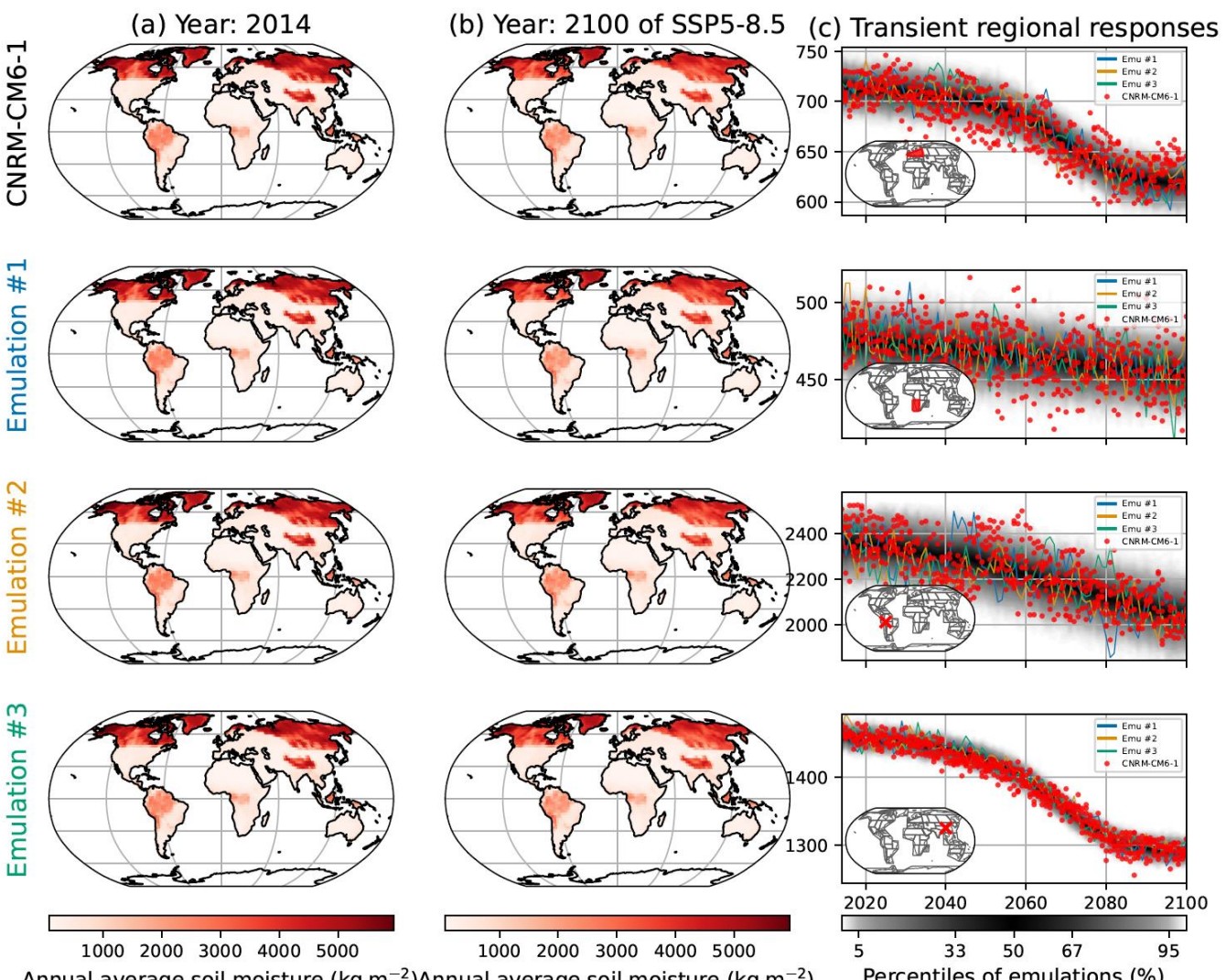

**Figure 9: Similar to Figure 2, although for the annual average soil moisture ($SM$) under CNRM-CM6-1.** The rows correspond from top to bottom to the West & Central Europe, the West of South Africa, a grid point in the west of Brazil in Acre and a grid point in Sichuan close to Chengdu.

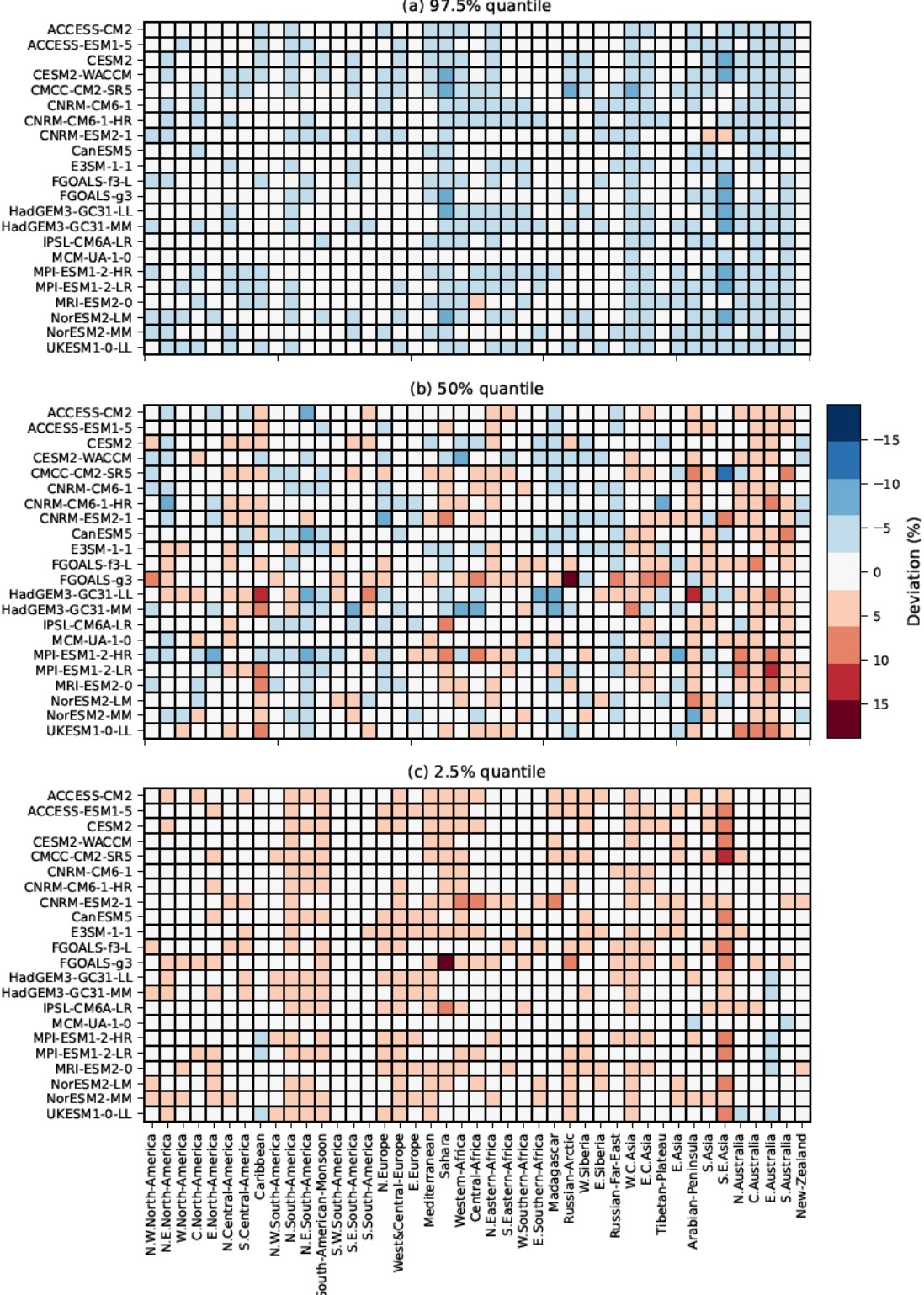

**Figure 10: Similar to Figure 3, although for the annual average soil moisture (*SM*).**

### 4.3 Emulation of the annual minimum of the monthly average of soil moisture

Emulating the annual minimum of the monthly average soil moisture is analogue to the emulation of annual average soil moisture. As an average over a month, *SMmm* may be represented using a normal distribution, although as the minimum over the months, it may be represented by a GEV distribution. Though, sampling a block-maxima over 12 values, the months, is too small to converge towards a GEV distribution. Thus, a normal distribution is used. Checking the local evolutions of the sample leads to

similar observations than observed for the annual average of the soil moisture illustrated in Figure 7. Thus, the same configurations are used for *SMmm* than for *SM*.

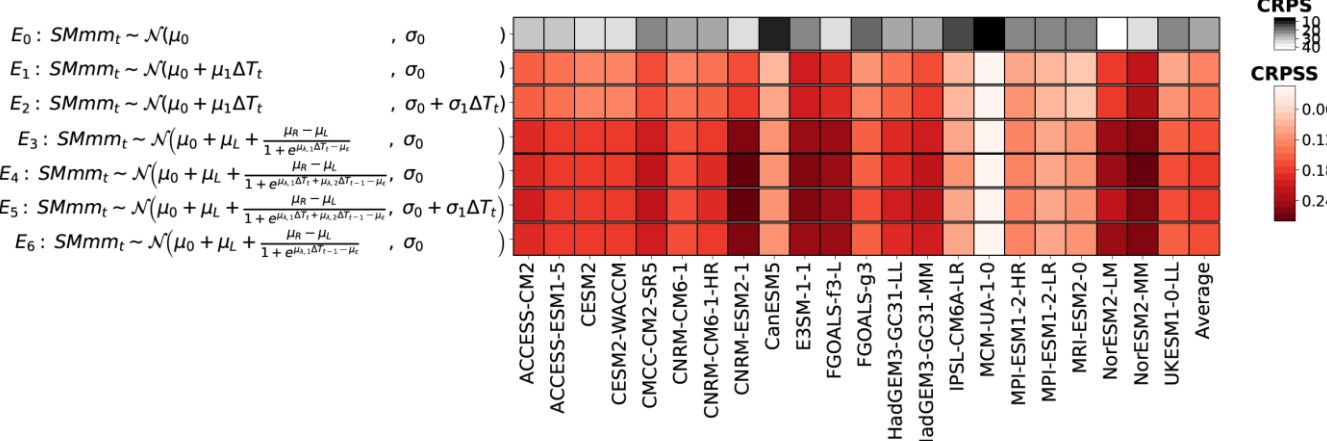

**Figure 11: Similar to Figure 1, although for the annual minimum of the monthly average of soil**
**moisture (*SMmm*).**

We summarize in Figure 11 the performances for the emulations of ***SMmm*** over the different configurations, with the coefficients **μ** and **σ** corresponding respectively to the location and the scale of the normal distribution. The configuration with the best performances is $E_4$, with the mean as a logistic function of **ΔT** at the year and the former year, while the standard deviation remains constant.

$$E_4: SMmm_{s,t} \sim \mathcal{N}\left(\mu_{s,0} + \mu_{s,L} + \frac{\mu_{s,R} - \mu_{s,L}}{1 + exp(\lambda_{s,1}\Delta T_t + \lambda_{s,2}\Delta T_{t-1} - \mu_{s,\varepsilon})}, \sigma_{s,0}\right) \quad (13)$$

Note that both **SM** and **SMmm** have the same best configuration. Both annual indicators are averages and **SMmm** has for upper limit **SM**, which may explain this result. We also note that ACCESS-CM2 shows better performances with a linear evolution of the standard deviation, though the opposite occurs with NorESM2-LM. Without logistic evolution, we note lower performances for high warming scenarios, because linear fits fail at reproducing the non-linear evolutions at high **ΔT**. Without **ΔT** at the former year, the performances of the emulations are reduced for low warming scenarios, because the water cycle get more time to stabilize to the current regime.

The results for the emulations of $SMmm$ under this configuration are illustrated in Figure 12. The spatial patterns of the ESM shown here on the top row, CNRM-CM6-1, are correctly reproduced by the emulations on the three following rows. The right column shows that the regional responses are correctly reproduced, with a majority of the ESM points being within the range of the emulations. Their dispersions seem to respect the distribution of the emulation, as will be confirmed with the regional performances in Figure 13. Just like $SM$, the realizations by CNRM-CM6-1 in the grid point in Brazil on the third row of column (c) shows a decrease in $SMmm$ over 2020-2050, then an increase over 2050-2060, then a decrease over 2060-2100. In the meantime, the emulations fail to reproduce these evolutions, decreasing at a slower pace over 2020-2050 and not increasing over 2050-2060. The processes explaining for such evolutions are not reproduced by the emulator, and more research would be needed to integrate them. Figures similar to Figure 12 are provided in the Appendix 6.8 for low and mid warming scenarios.

The performances of the emulations for the retained configuration for $SMmm$ are shown in Figure 13. The deviations of quantiles of the emulations to the ESMs are summarized for each ESM and AR6 region respectively at the quantile 97.5%, 50% and 2.5%. The emulators are here again overall underdispersive. On average, the fraction of points above the 97.5% quantile of emulations indicate that this quantile of the emulations are too low by -2.0%. At the median, the emulations are +1.1% too high. At the 2.5% quantile, the emulations are +1.4% too high. The fraction of ESMs x regions with a deviation of quantiles limited to 5% is limited to 96% for both 97.5% and 2.5% quantiles and at 85% for the median. Overall,

the distributions are relatively well reproduced, although some regions show lower performances. Here again, the emulator performs lower in South-East Asia than in the other regions. As explained in other sections, this may be an effect of less land grid points affecting the reproduction of spatial correlations.

On the median, the emulator of MCM-UA-1 has lower performances than for the other ESMs. The emulator of NorESM2-LM has lower performances on the two other shown quantiles. These results cannot be used directly to diagnose different effects in the ESMs. Instead, further research will be needed to understand and integrate these effects in the modelling framework of MESMER-X.

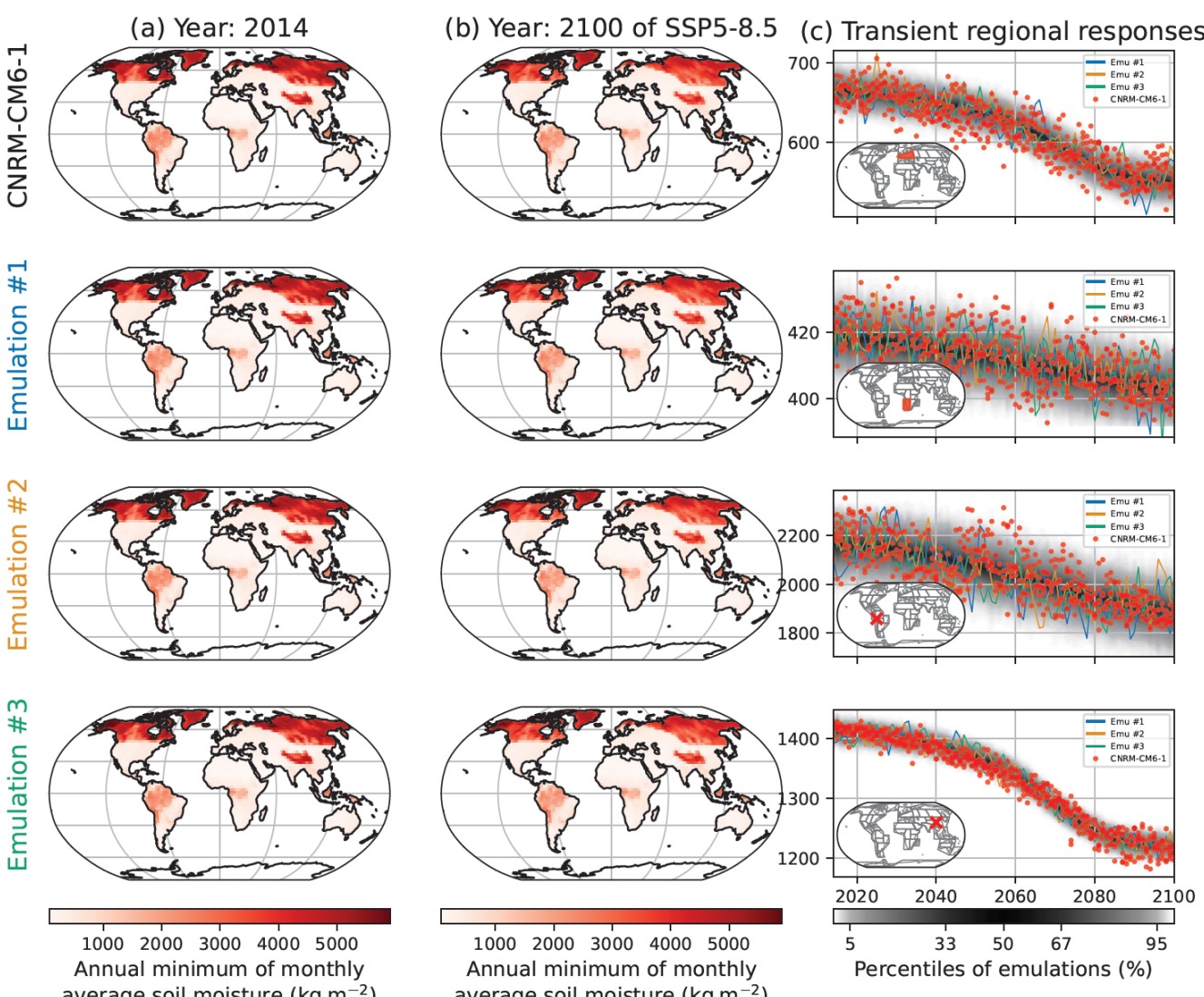

**Figure 12: Similar to Figure 2, although for the annual minimum of the monthly average of soil moisture (*SMmm*) under CNRM-CM6-1.** The rows correspond from top to bottom to the West & Central Europe, the West of South Africa, a grid point in the west of Brazil in Acre and a grid point in Sichuan close to Chengdu.

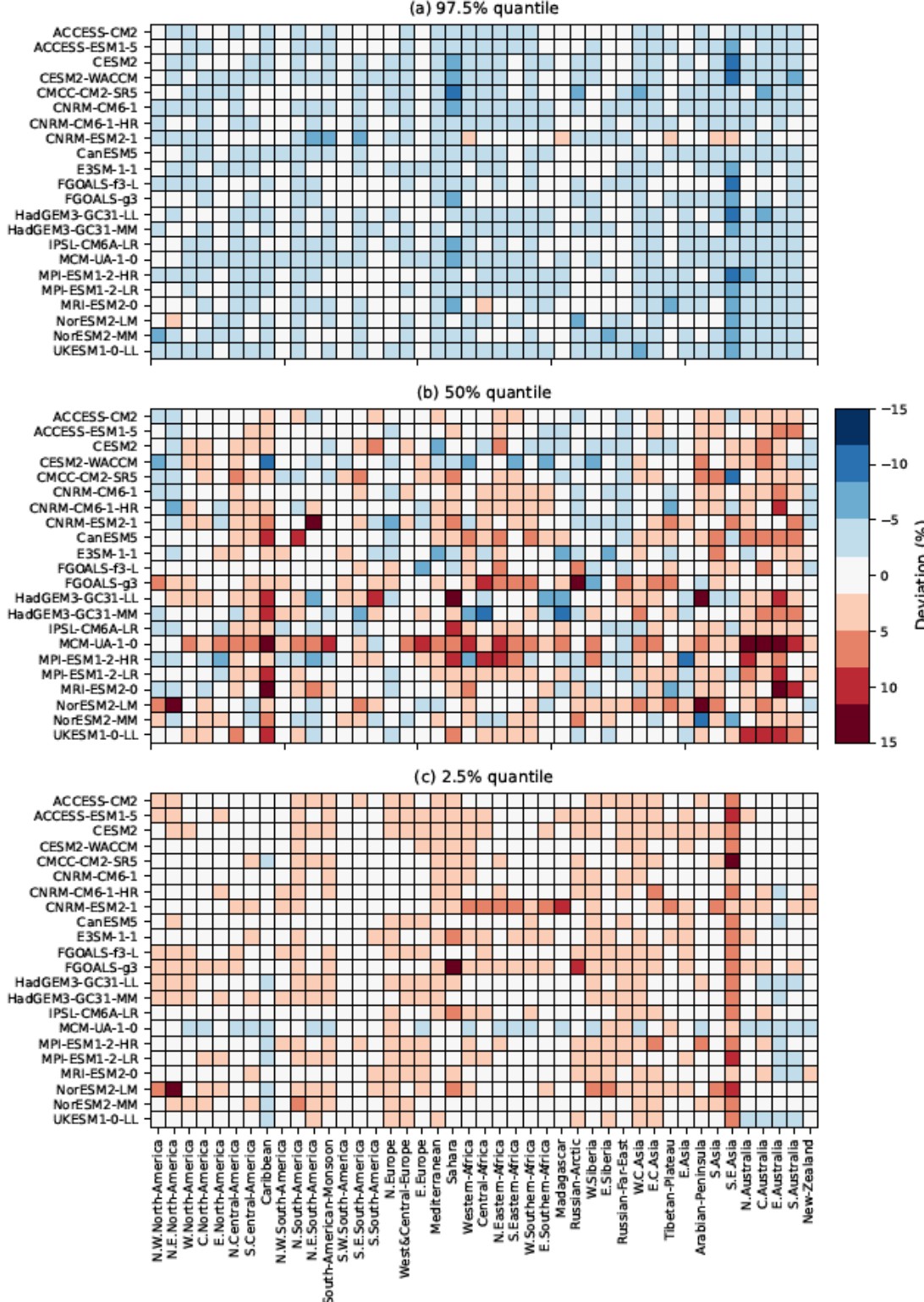

**Figure 13: Similar to Figure 3, although for the annual minimum of the monthly average of soil moisture (*SMmm*).**

## 5 Conclusions

The emulator MESMER-X, an extension of the MESMER emulator (Beusch et al., 2020a; Beusch et al., 2022b) which is focused on the emulation of impact-relevant variables, including extremes, was introduced and showcased for TXx (Quilcaille et al., 2022), suggesting a potential for extension to other climate variables. Here, we have confirmed this potential with a range of yearly indicators of the fire weather index and soil moisture. We illustrated that several distributions may be used in this framework, such as the GEV for TXx and FWIsa, the normal distribution for SM and SMmm and finally the Poisson distribution for FWIxd. It clearly shows how the MESMER-X framework can be easily adapted to sample from additional probability distribution, thereby facilitating its adaptation to further climate variables. Moreover, the non-linear response of soil moisture to global mean temperature required a more sophisticated parameterization, including a logistic response and the consideration of time-lagged predictor variables. This latter extension highlights that the MESMER-X setup can be easily adapted to also account for a non-linear climate response in the considered variable.

We have shown good performances for these emulators, typically with deviation on quantiles limited to 5% in about 90% of the ESMs x AR6 regions, with variations on the indicators and quantiles. We have pointed out some limitations. The main one was observed with FWIxd, with lower performances on the median of emulations. In this case, the Poisson distribution may not be adequate, more flexibility in the moments of the distribution may be necessary for instance to allow fat tails. Another limitation is that there are regions that would benefit from local responses with different parametrizations, e.g. with fire indicators in South America. Such effects have not been accounted for here, to preserve simplicity in the modeling. Making parametrizations dependent on the grid point would be a solution but wasn't implemented for this article. Finally, some local aspects of the dynamics are not captured by the emulations, e.g. with soil moisture indicators in Amazonia. Using time-lagged predictors may be not good enough locally, or there may even be processes that cannot be entirely captured in this framework.

Given these results, the further expanded MESMER-X emulator is capable of emulating several annual impact-related variables, including climate extremes and a drought-related water-cycle variable, with satisfactory performances. It can emulate variables distributed over GEV, normal and Poisson distribution. Linear, quadratic and logistic evolutions on the parameters have been shown here. An example of lagged effect is shown here. This method is very flexible, relatively simple, and yet has good performances. We have identified limitations, but also proposed potential solutions.

The expanded MESMER-X is thus a tool now capable of exploring impact-related variables, including climate extremes and a drought-related water-cycle variable, and may be used to provide information to assess climate impacts under a range of emissions scenarios, also upcoming scenarios to be developed in preparation to the 7th Assessment report of the IPCC. As such, the MESMER-X emulator is complementary to the ESMs: it relies on ESMs for training but is fast enough for coupling with other models in need of climate information. Finally, ESMs may carry some biases (Kim et al., 2020), even on climate extremes (Schewe et al., 2019). Tools such as MESMER-X may foster the integration of observations constraints to correct these biases.

## 6 Appendices

### 6.1 Application of a Probability Integral Transform to discrete distributions

The Probability Integral Transform (PIT) introduced in Equation (2) of the manuscript transforms values from a known distribution to another distribution, here a normal distribution of mean 0 and standard deviation 1, thus "gaussianising" the sample. We illustrate here how the PIT applies to discrete distributions. For the sake of clarity, these explanations are not based solely on statistical data instead of climate data.

We consider here a GEV distribution and a Poisson distribution. To facilitate the comparison, the parameters are picked so that their cumulative distribution functions (CDFs) would be relatively similar. We show in Figure A. 1 their respective CDFs, how the PIT would apply to two values.

We note that events with a value of 4 would have higher transformed values under a Poisson distribution than under a GEV distribution. This observation may raise issues regarding the use of a PIT for a discrete

distribution. However, we remind that a value of 4 is representative of the values in the interval [3.5; 4.5[.
Thus, over [3.5; 4[, the transformed values over a Poisson distribution would be below those of a GEV,
while over [4; 4.5[, they would be higher than those of a GEV. According to this effect, applying a PIT
to a discrete distribution would lead to partially compensating errors.

Intervals from the discrete distribution are represented by a single value, thus a single value in the
"gaussianised" space. However, the realizations from the auto-regressive process with spatially correlated
innovations are back-transformed using another PIT, as described in Equation (7). These realizations are
continuous, not taking only the values taken by a Poisson distribution after PIT. As such, the same effect
occurs, though in the other way round: intervals of values in the realizations are transformed into single
values.

As a result, applying a PIT to a discrete distribution appears to have the intended effect. This is due to the
matching of intervals of values to single values, which lead to partially compensating effects.
Furthermore, this effect occurs another time during the back transformation. We acknowledge the extent
of the compensations of these effects, will investigate further in this direction and welcome other
contributions.

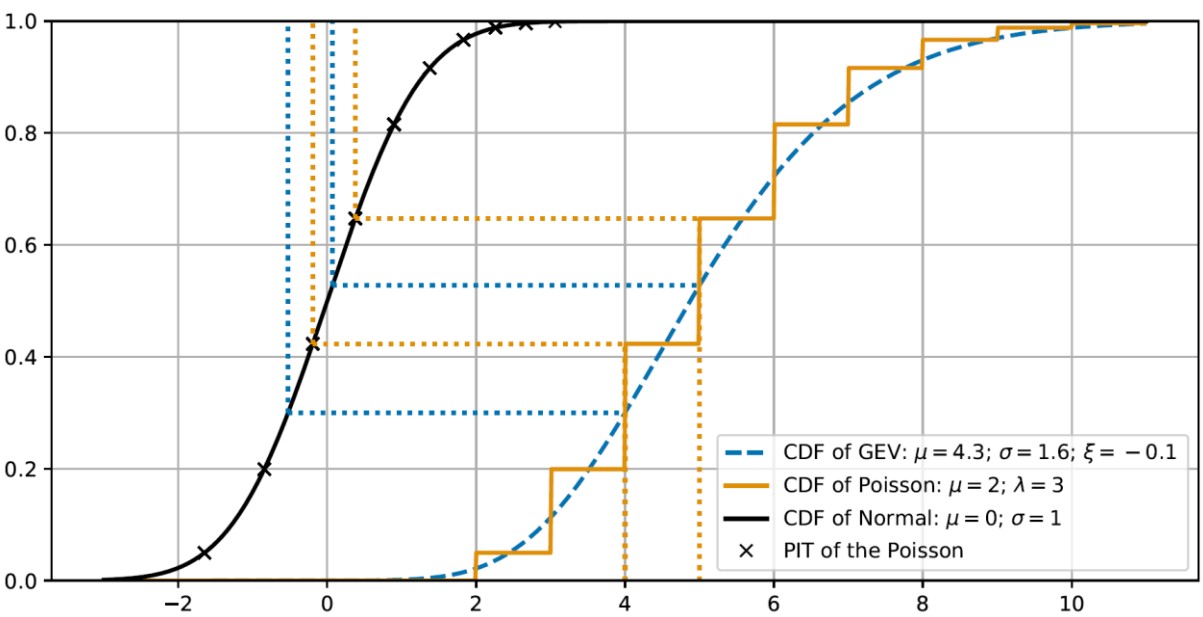

**Figure A. 1: Illustration of a Probability Integral Transform applied to continuous and discrete distributions.**

## 6.2 Representation of the interannual variability for each variable

An important aspect of the impacts of climate change is their potential persistency. Hazardous climate conditions impact the Earth system and our societies, but such conditions maintained over several years may result in even higher impacts. For instance, droughts lasting several years would have stronger impacts in terms of food security than the impacts of non-adjacent droughts.

As such, representing the interannual variability matters when emulating variables related to climate impacts or the water cycle. In MESMER-X, it is modeled using an auto-regressive process of 1$^{st}$ order, as shown in equation (3). It is applied on the climate variable after the probability integral transform of equation (2), to ensure a "gaussianised" distribution, required by the auto-regressive process. However, the training of this process is performed over the whole training sample, and the interannual variability of the ESM may change over time, for instance due to changes in large scale oscillations.

Here, we evaluate the local evolutions of the interannual variability in the trained ESMs and is representation by MESMER-X. For each climate variable emulated in this paper, we use the ESM used for illustration of its emulator in time series and maps. We choose three periods, the preindustrial (1851-1900), the end (2051-2100) of a low warming scenario SSP1-2.6 and the end (2051-2100) of a high warming scenario SSP5-8.5. In each case, we apply the probability integral transform as shown in equation (2), as a form of detrending and so that the new sample follows a standard normal distribution. In each grid point, we calculate an auto-regressive process of 1$^{st}$ order, and average its coefficient over available members. For the emulator, we verify that these calculation effectively lead to the parameters $\gamma_{s,1}$ of equation (3), because the spatially correlated innovations over the realizations. All these results are shown in Figures A.15 to A.18.

These figures show that all the variables presented in this article are mostly positively correlated. Besides, $SM$ and $SMmm$ have higher correlations than $FWIsa$ and $FWIxd$. This is due to inertias in the water cycle, with relatively long recovery time from droughts. The evolutions of these correlations in the ESM are relatively slow, mostly in Québec, Greenland and in Murmansk. Its MESMER-X counterpart is the average in time of these correlations, thus reproducing well the interannual variability of the ESM.

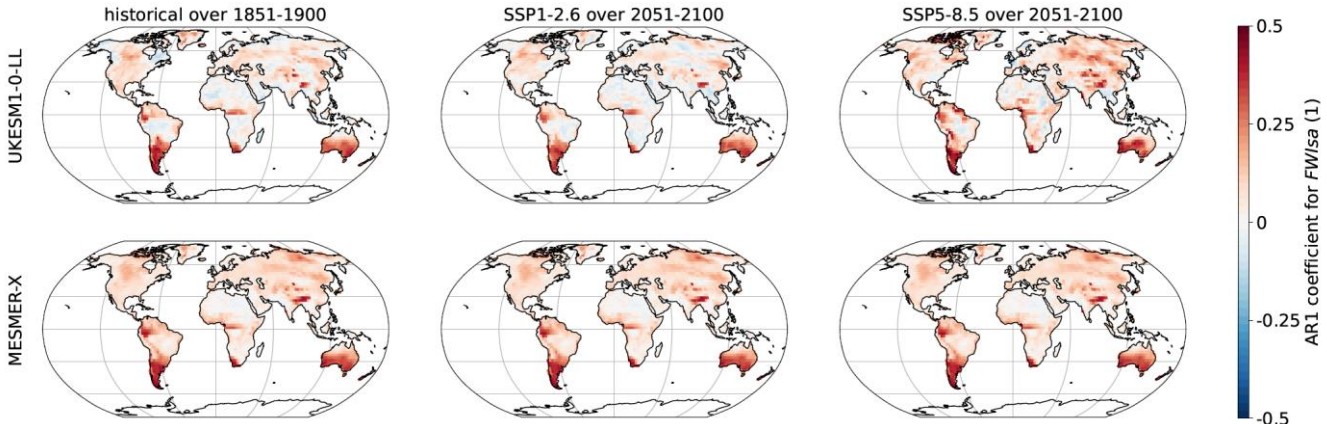

**Figure A. 2: First order coefficient of a temporal auto-regressive process for $FWIsa$ with UKESM1-0-LL and MESMER-X using the configuration presented in equation (10).**

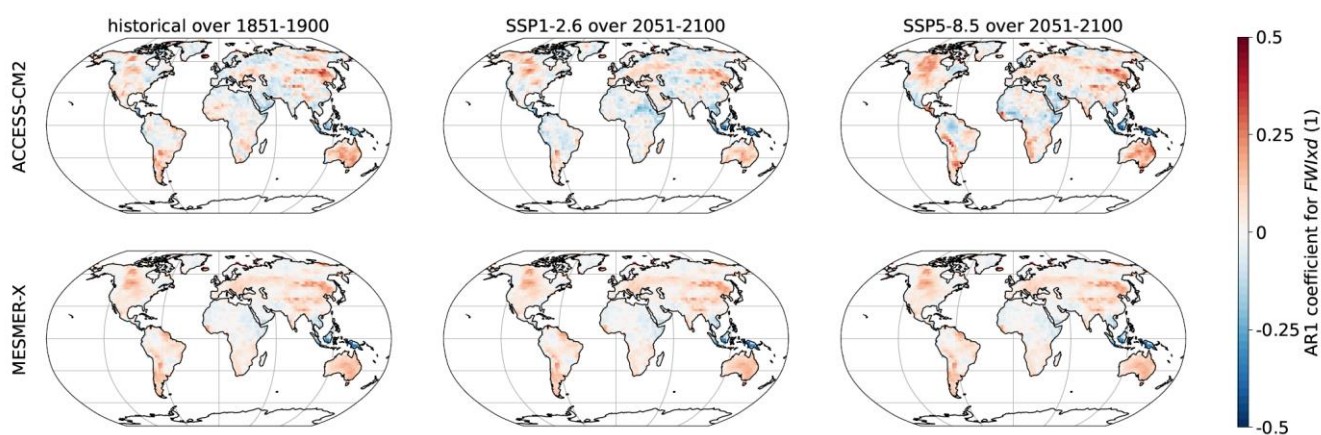

**Figure A. 3: First order coefficient of a temporal auto-regressive process for $FWIxd$ with ACCESS-CM2 and MESMER-X using the configuration presented in equation (11).**

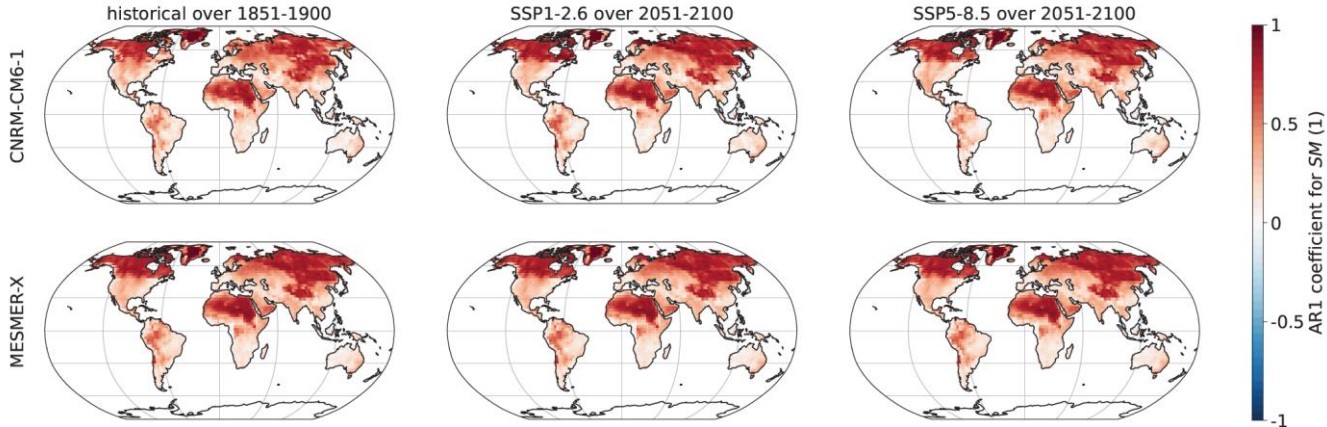

**Figure A. 4: First order coefficient of a temporal auto-regressive process for *SM* with CNRM-CM6-1 and MESMER-X using the configuration presented in equation (12).**

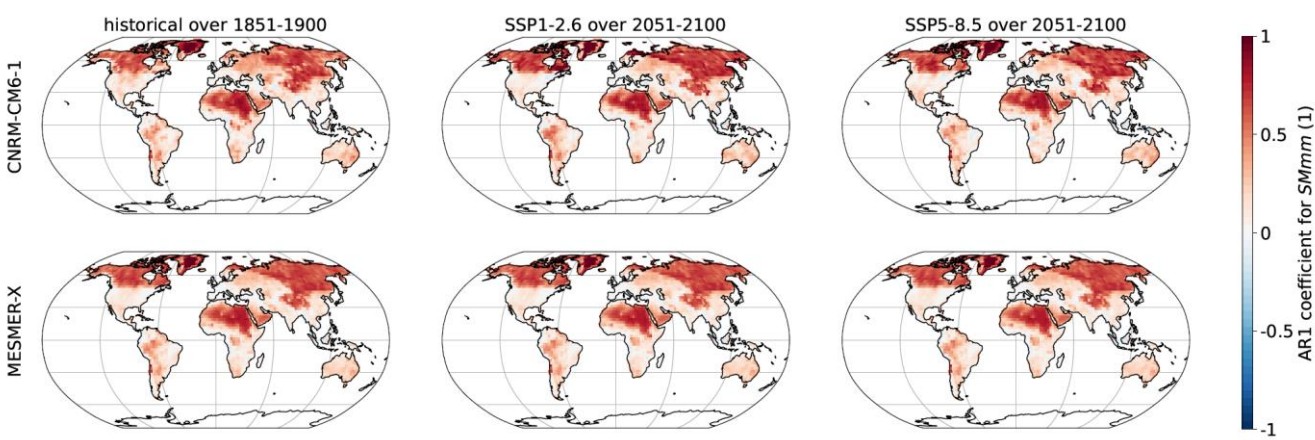

**Figure A. 5: First order coefficient of a temporal auto-regressive process for *SMmm* with CNRM-CM6-1 and MESMER-X using the configuration presented in equation (13).**

## 6.3 Interpretability of the CRPS

All CRPS scores of this manuscript have been calculated thanks to the Python package *properscoring* available at https://pypi.org/project/properscoring/, more specifically with its function calculating *crps_ensemble*. Below is an illustration of the CRPS obtained using this function.

For interpretability of the CRPS, one may consider the expression for an observation X and the Normal distribution, with $f$ and $\mathcal{F}$ respectively its probability density and cumulative distribution functions, derived from the equation 8.55, p. 353 of (Wilks, 2011):

$$CRPS = \sigma \left( X(2\,\mathcal{F}_{\mathcal{N}}(X,\mu,\sigma) - 1) + 2f_{\mathcal{N}}(X,\mu,\sigma) - {1}/{\sqrt{\pi}} \right)$$

Similar equations may be obtained for a GEV from equation 9 of (Friederichs and Thorarinsdottir, 2012), or other distributions (http://cran.nexr.com/web/packages/scoringRules/vignettes/crpsformulas.html).

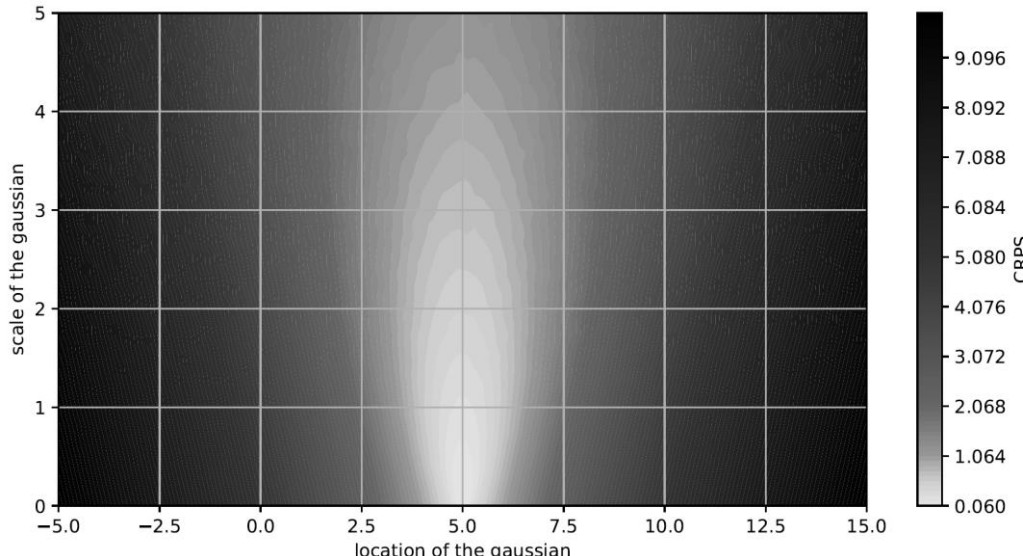

**Figure A. 6: CRPS obtained with an observed value of 5 and gaussian distributions sampled over 10.000 members over different values of its parameters.**

## 6.4 Performances of the emulators for each variable

The grid-cell level parameters of MESMER-X are trained by minimizing the negative log-likelihood of the training sample given a prescribed configuration for each grid-cell independently. We show here the averaged negative log-likelihood at the grid cell level for the retained configuration and with the ESM used to illustrate the performances of MESMER-X. The value is averaged to account for the number of time steps used during training and facilitate the comparisons.

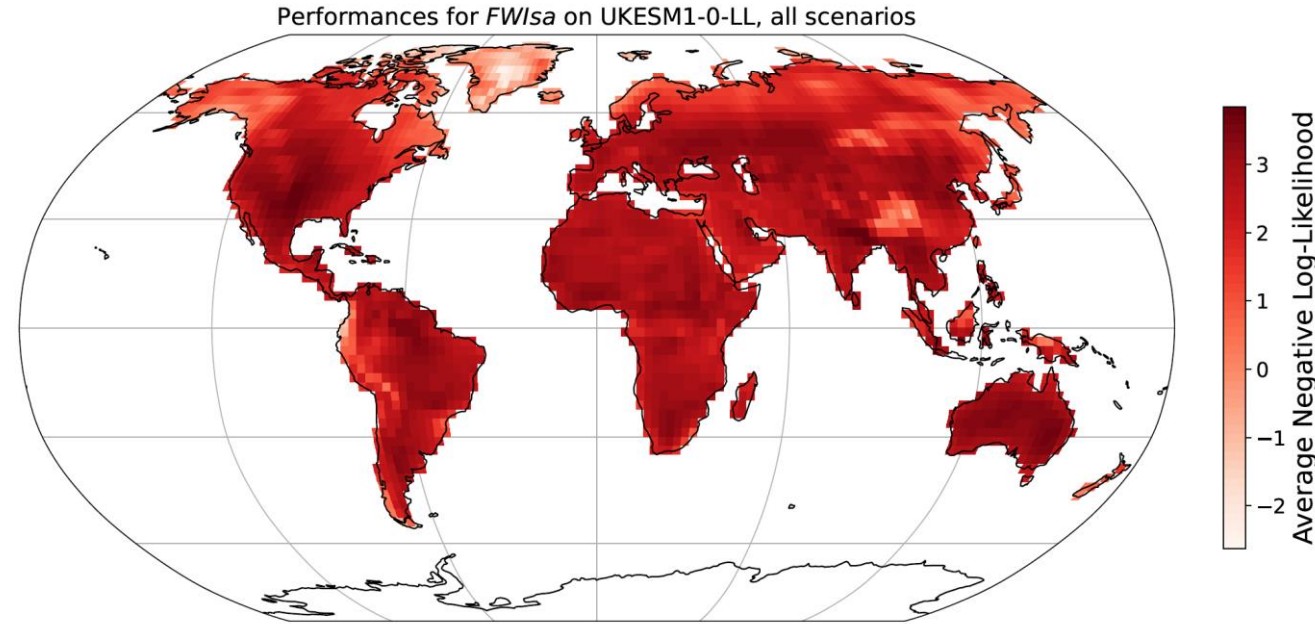

**Figure A. 7: Negative log-likelihood obtained during training of MESMER-X on $FWIsa$ using the configuration presented in equation (10) and for the ESM used in Figure 2.**

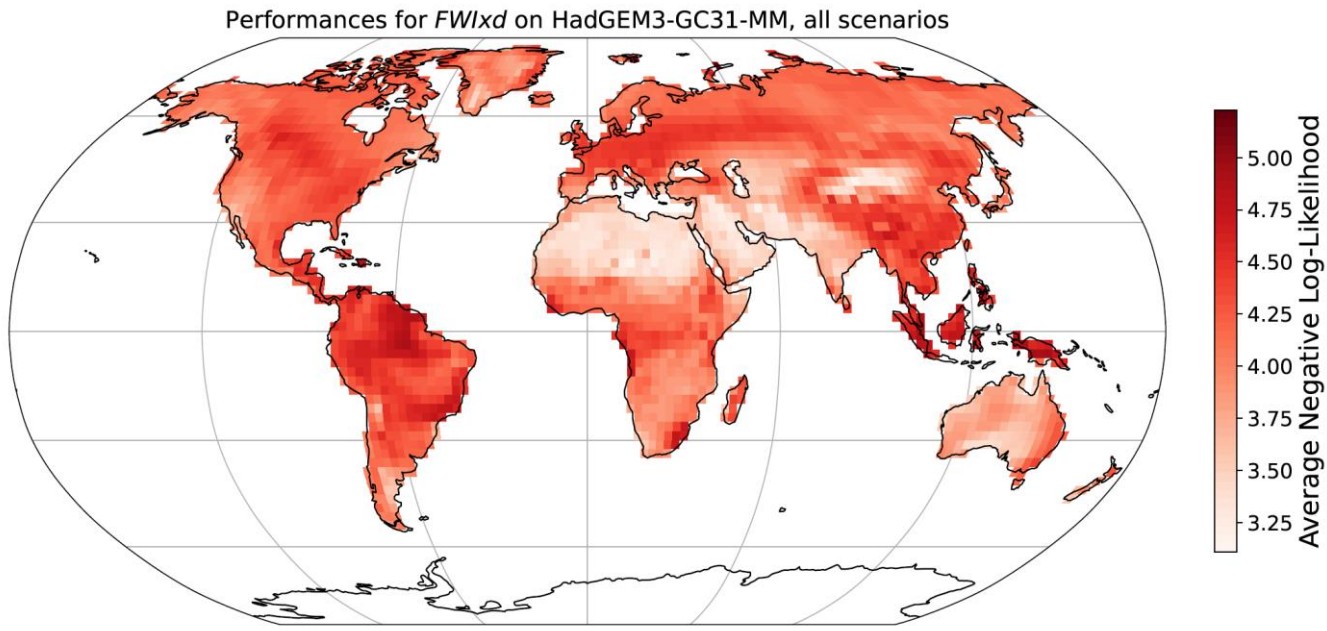

**Figure A. 8: Negative log-likelihood obtained during training of MESMER-X on $FWIxd$ using the**
680 **configuration presented in equation (11) and for the ESM used in Figure 5.**

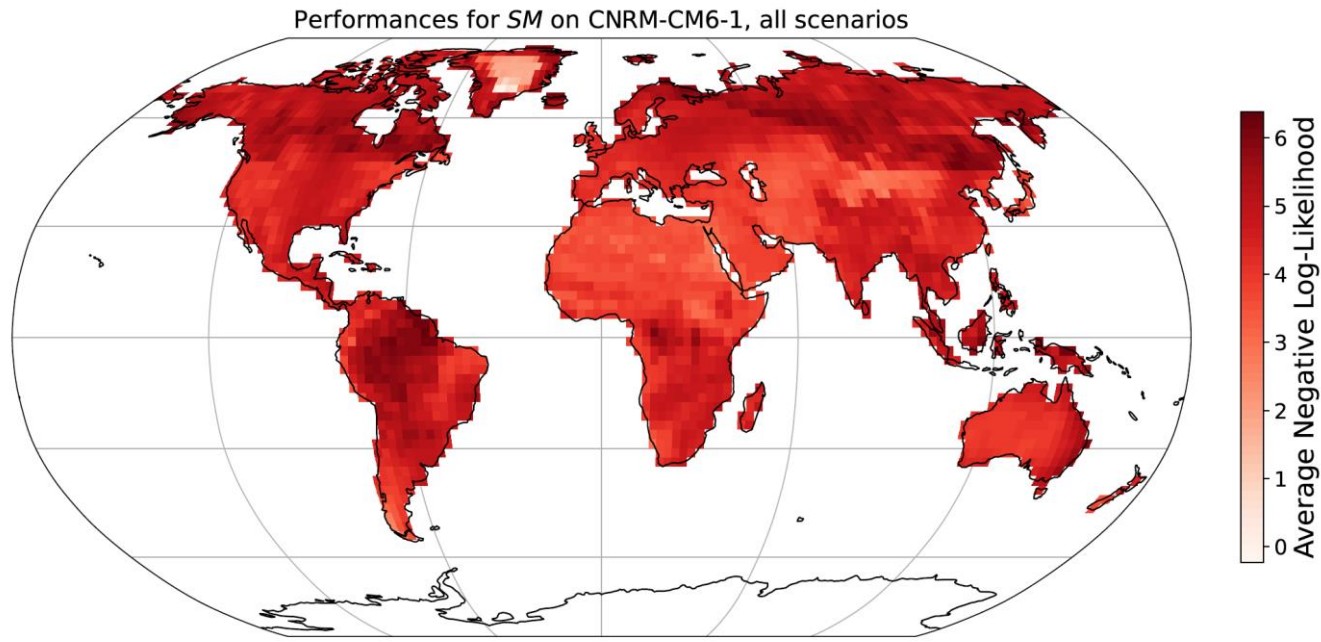

**Figure A. 9: Negative log-likelihood obtained during training of MESMER-X on *SM* using the configuration presented in equation (12) and for the ESM used in Figure 9.**

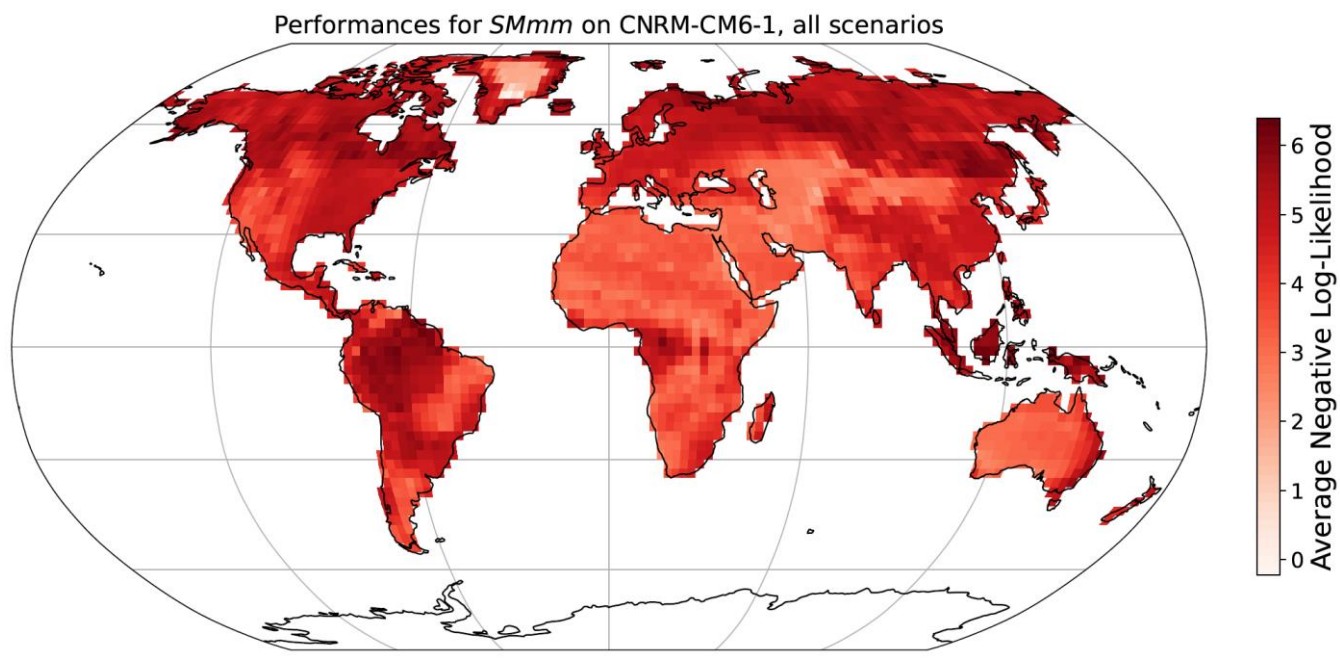

**Figure A. 10: Negative log-likelihood obtained during training of MESMER-X on *SMmm* using the configuration presented in equation (13) and for the ESM used in Figure 12.**

## 6.5 Emulations of the seasonal average of the Fire Weather Index over low and mid warming scenarios

In Section 3.2, we emulate the seasonal average of the Fire Weather Index ($FWIsa$), that we illustrate in Figure 2 with the high warming scenario SSP5-8.5. While this scenario allows to explore a large range of warming for the model, it does not show evolutions over more advisable warming ranges, nor does it show potential stabilisation effects over low warming scenarios. Here, we produce the equivalent of Figure 2 for SSP1-2.6 and SSP2-4.5.

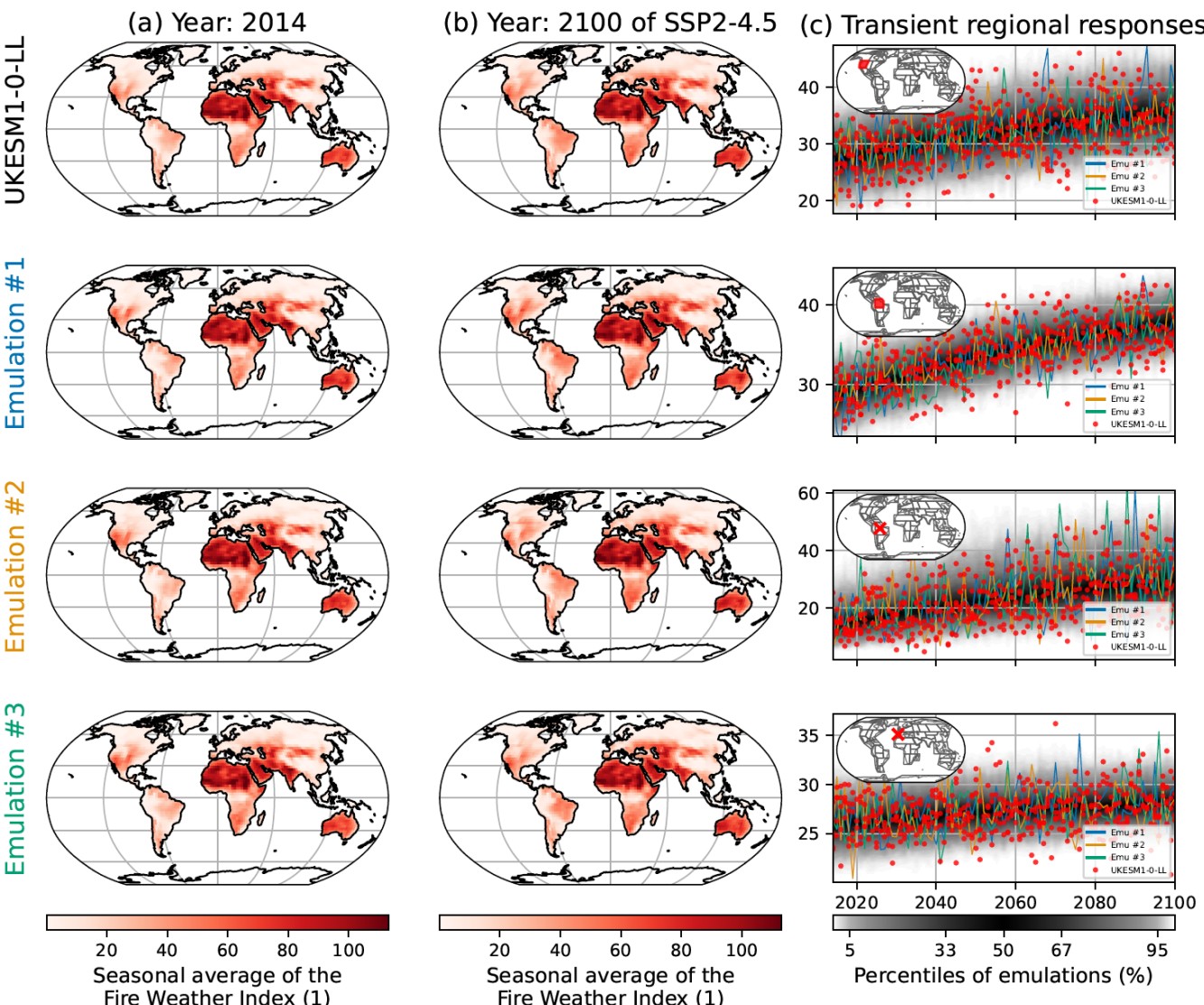

**Figure A. 11:** Similar to Figure 2, although with the mid warming scenario SSP2-4.5.

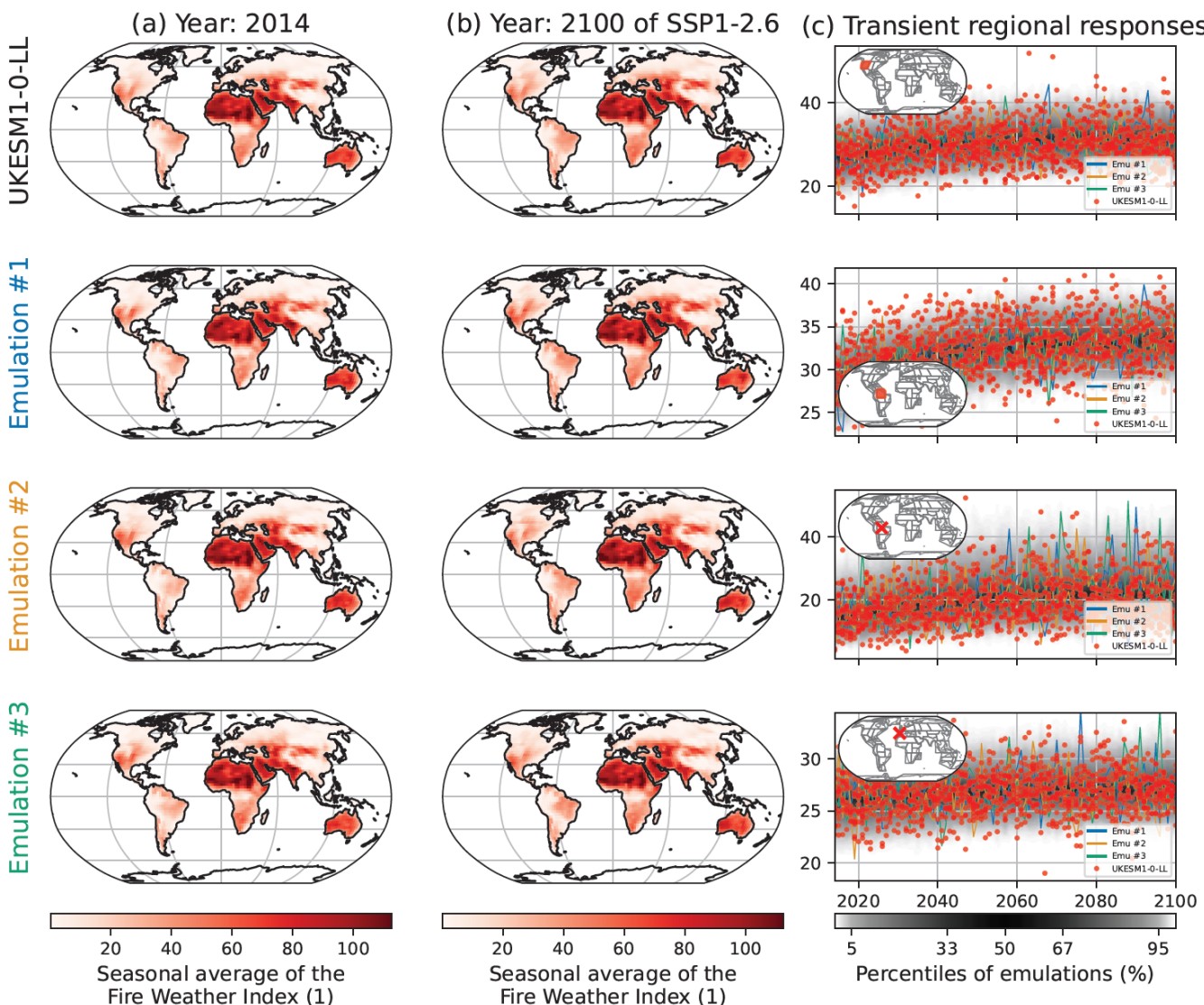

**Figure A. 12: Similar to Figure 2, although with the low warming scenario SSP1-2.6.**

## 6.6 Emulations of the number of days with extreme fire weather over low and mid warming scenarios

Like done in Section 6.3 for the seasonal average of the Fire Weather Index, we extend Section 3.3, where we emulated the number of days with extreme fire weather ($FWIxd$) and illustrated in Figure 5 with the high warming scenario SSP5-8.5. Again, while this scenario allows to explore a large range of warming for the model, it does not show evolutions over more advisable warming ranges, nor does it show potential stabilisation effects over low warming scenarios. Here, we produce the equivalent of Figure 5 for SSP1-

 2.6 and SSP2-4.5. We highlight that the SSP2-4.5 was not provided by the ESM HadGEM3-GC31-MM. Also, its counterpart HadGEM3-GC31-LL provided only one member for SSP1-2.6. For the sake of visualisation, we opt to show the results with ACCESS-CM2 which provided 5 members for both SSP1-2.6 and SSP2-4.5.

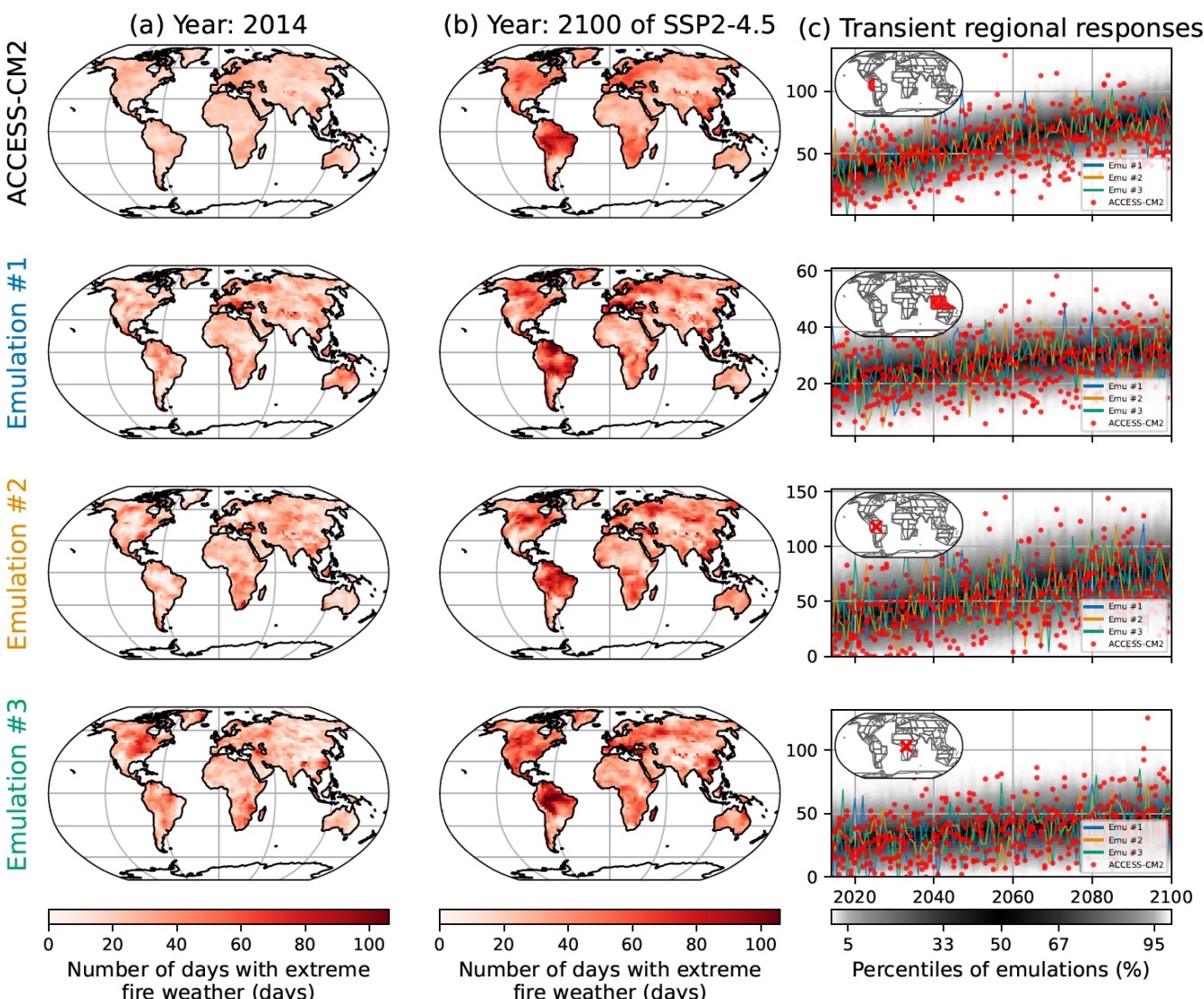

 **Figure A. 13:** Similar to Figure 5, although with ACCESS-CM2 and the mid warming scenario SSP2-4.5.

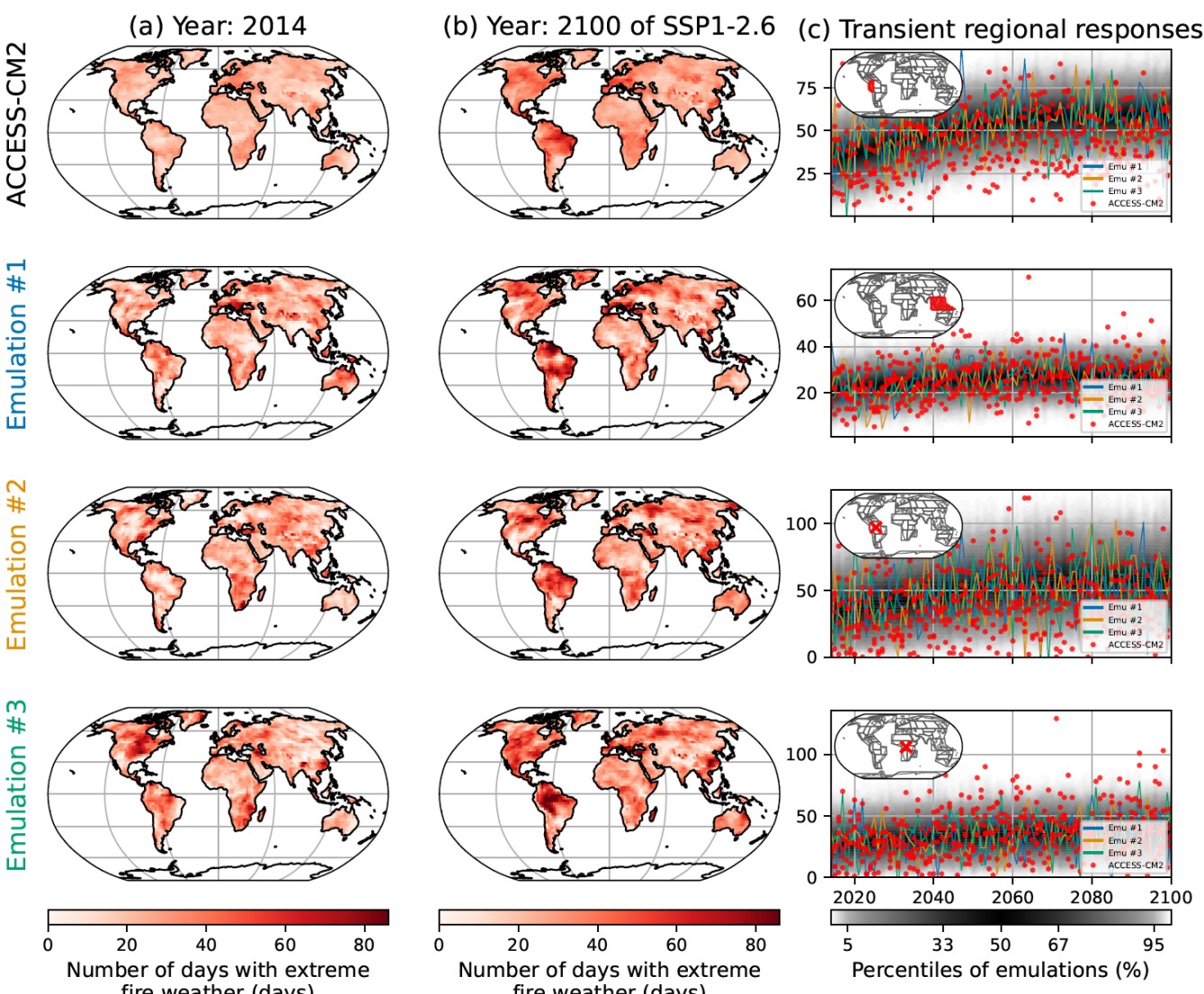

**Figure A. 14:** Similar to Figure 5, although with ACCESS-CM2 and the low warming scenario SSP1-2.6.

715

## 6.7 Emulations of the annual average of the soil moisture over low and mid warming scenarios

Like done in Section 6.3 for the seasonal average of the Fire Weather Index, we extend Section 4.2, where we emulated the annual average of the soil moisture ($SM$) and illustrated in Figure 9 with the high warming scenario SSP5-8.5. Again, while this scenario allows to explore a large range of warming for

the model, it does not show evolutions over more advisable warming ranges, nor does it show potential stabilisation effects over low warming scenarios. Here, we produce the equivalent of Figure 9 for SSP1-2.6 and SSP2-4.5.

In Figure A. 16, the time series on the last row show that the emulations are more optimistic than the ESM in this grid point from 2080. A potential explanation would be that the effect introduced by lagged temperatures becomes too strong. As outlined in this article, different parametrizations of the inertias in the water cycle may improve the representation of such local effects or having parametrizations depending on the grid point instead of being identical for all of them.

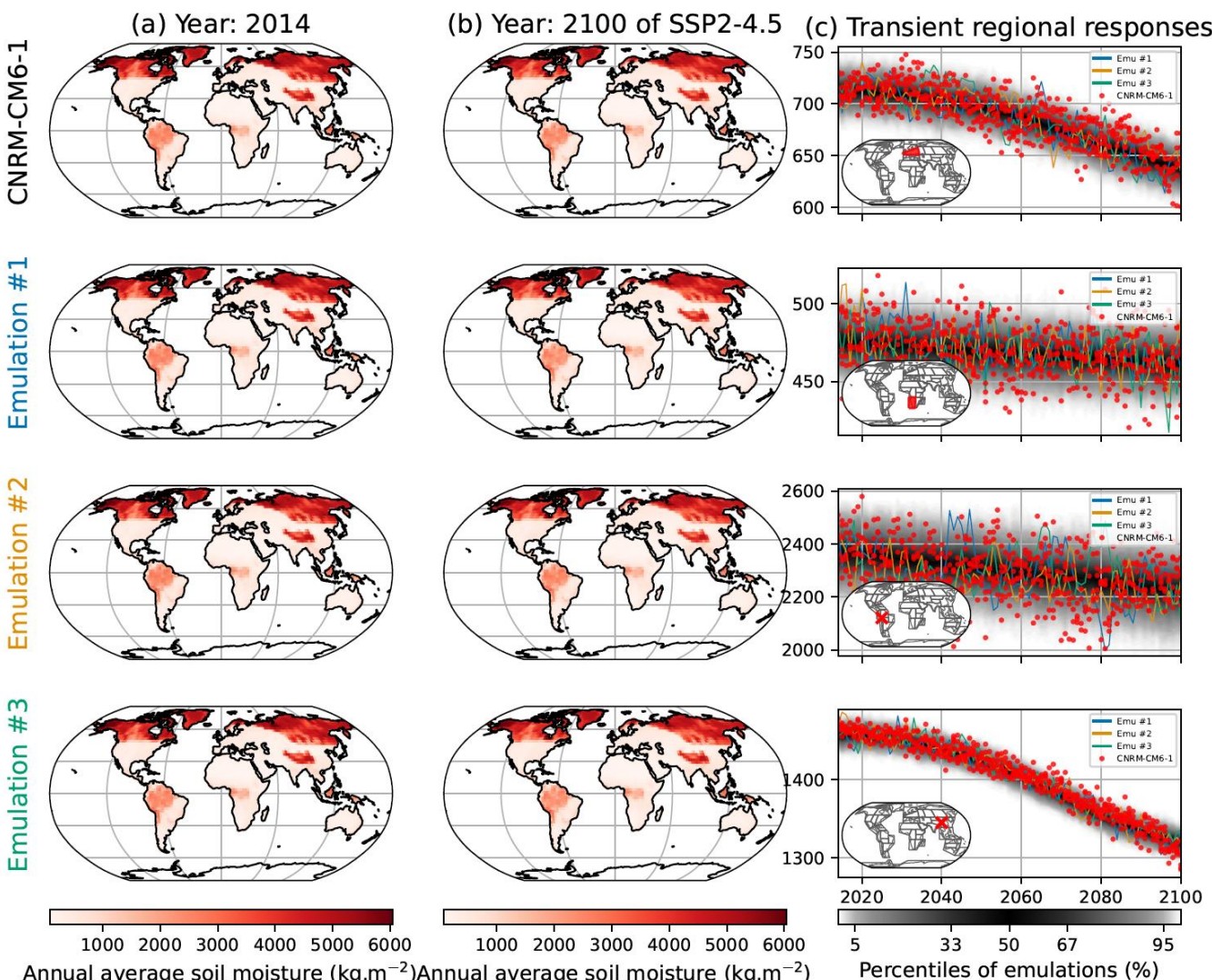

**Figure A. 15: Similar to Figure 9, although with the mid warming scenario SSP2-4.5.**

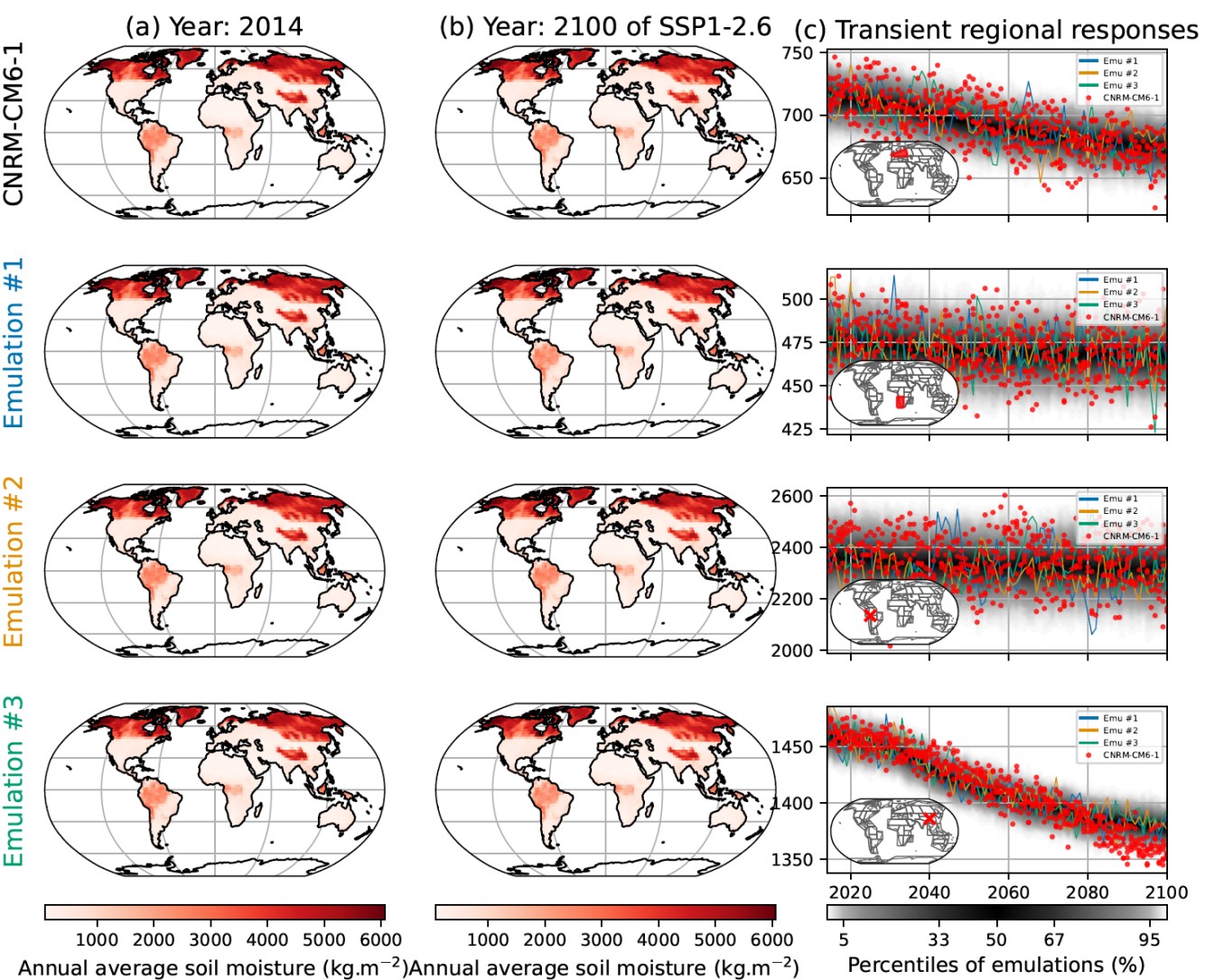

**Figure A. 16: Similar to Figure 9, although with the low warming scenario SSP1-2.6.**

## 6.8 Emulations of the annual minimum of the monthly average soil moisture over low and mid warming scenarios

Like done in Section 6.3 for the seasonal average of the Fire Weather Index, we extend Section 4.3, where we emulated the annual average of the soil moisture ($SMmm$) and illustrated in Figure 12 with the high warming scenario SSP5-8.5. Again, while this scenario allows to explore a large range of warming for the model, it does not show evolutions over more advisable warming ranges, nor does it show potential

stabilisation effects over low warming scenarios. Here, we produce the equivalent of Figure 12 for SSP1-2.6 and SSP2-4.5.

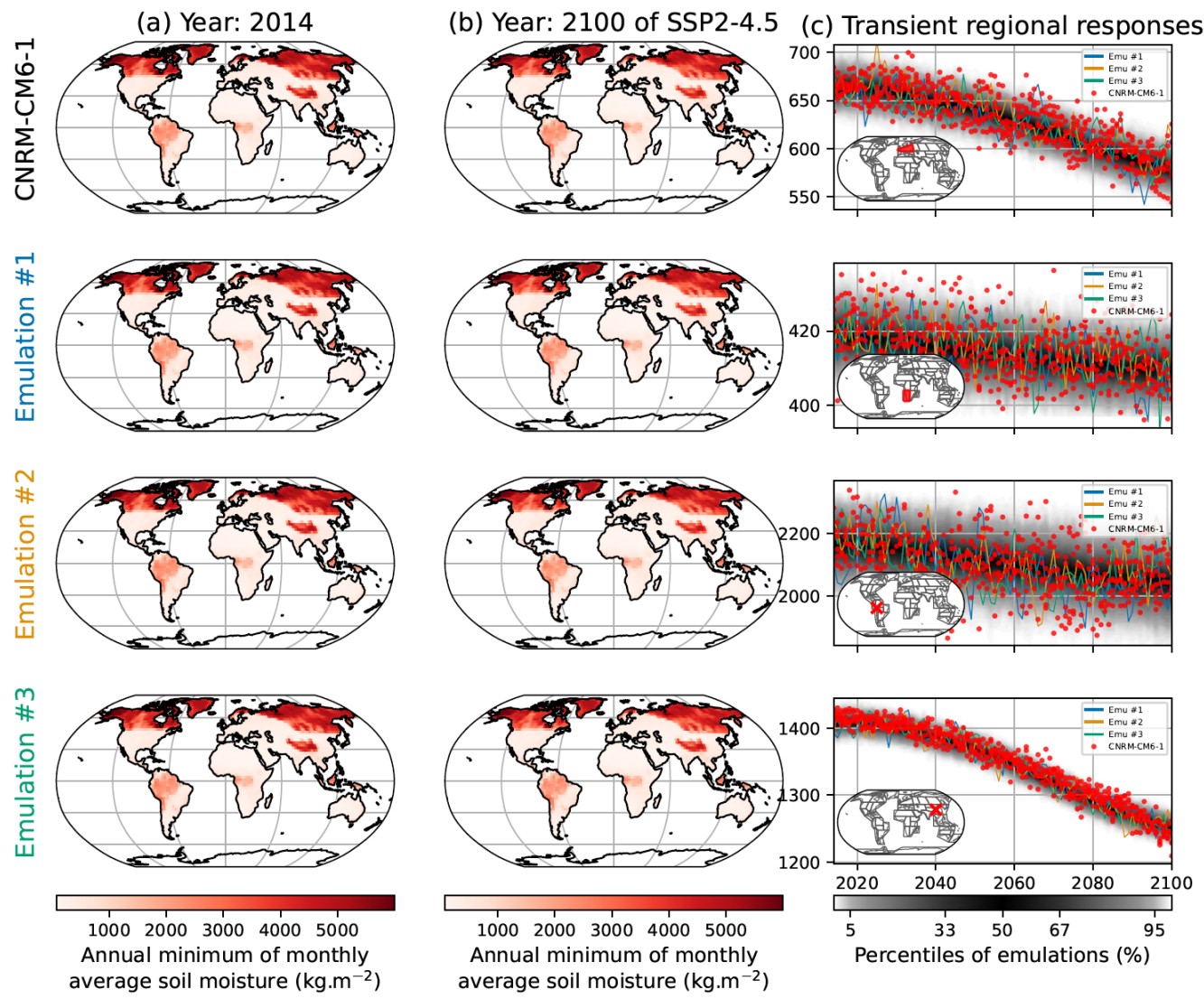

Figure A. 17: Similar to Figure 12, although with the mid warming scenario SSP2-4.5.

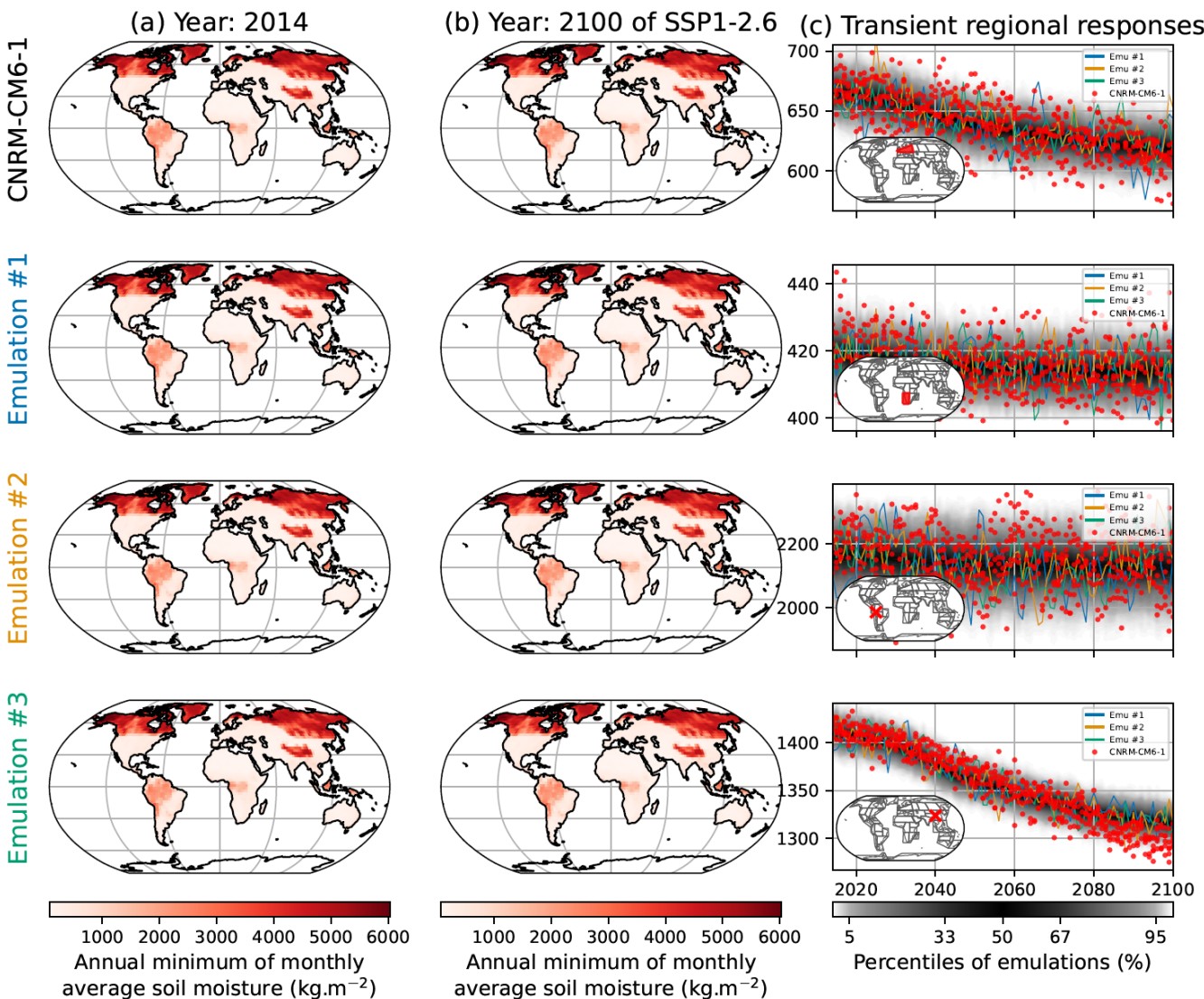

**Figure A. 18: Similar to Figure 12, although with the low warming scenario SSP1-2.6.**

**Acknowledgments**

We acknowledge the World Climate Research Program for the promotion and coordination of the exercise and data CMIP6, but also the climate modeling groups for their model outputs. We acknowledge the Earth System Grid Federation for archiving the data and providing access. We thank Urs Beyerle for downloading and curating the CMIP6 data. We thank Lukas Brunner for the processing into the CMIP6-ng archive. We acknowledge funding from the European Research Council (ERC) through the ERC

Proof-Of-Concept Grant MESMER-X (Grant Agreement ID 964013) and the PROVIDE project (Grant Agreement ID 101003687).

## Open research

Data from CMIP6 can be accessed and downloaded at https://esgf-node.llnl.gov/search/cmip6/ (last
access on the 29 January 2023). The search query is as follows: Experiment ID (historical, ssp119, ssp1226, ssp245, ssp370, ssp585, ssp534-over) and variable (tas, mrso). Code from MESMER is available at https://github.com/MESMER-group/mesmer. Data for the FWI can be accessed at https://doi.org/10.3929/ethz-b-000583391.

## Competing interests

Yann Quilcaille, Lukas Gudmundsson and Sonia I. Seneviratne declare that they do not have any competing interests. Though, for the sake of transparency, we notify that Sonia I. Seneviratne is a member of the editorial board of Earth System Dynamics. However, we point out that it had no impact on the reviewing and editing processes of this manuscript, for which the handling editor is Vivek Arora.

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
