# Peer review of "Extending MESMER-X: A spatially resolved Earth system model emulator for fire weather and soil moisture"

_EGUsphere, 2023_

## Referee Comment (RC1)

Review of: Extending MESMER-X: A spatially resolved Earth system model emulator for fire weather and soil moisture by Quilcaille et al.

This paper details a modification/extension of the established MESMER-X approach to emulating spatially and temporally resolved impact-relevant variables. The targets of the emulations are historical and scenario outputs from CMIP6-era ESMs, focusing specifically on quantities relevant to wildfire risk, i.e., Fire Weather Index and soil moisture, and in particular measures of their tail behavior.

The paper is clearly written (maybe some occasional oddity due to non-native English usage, I know about that issue myself ☺), well organized and interesting. It shows the value and promise of the MESMER-X approach, highly relevant for integrating climate information and impact/mitigation analysis, and I think it will be publishable after some minor revision. A lot of what is presented is – as the title states – the extension of a methodological framework that has been established and peer reviewed already, so my comments do not question that part, but are mainly suggestions for further validation/clarification/expansion.

I start from perhaps the only significant request: I think it would be good to extend the validation in two directions: the first one is interannual variability, right now obfuscated by considering all realizations together and evaluating only the relative positions of target vs. emulated realizations in those plots showing many time series at once and their envelopes. Since interannual variability may be important in creating and making persistent some of these hazardous conditions, some simple metric that compares it between true and emulated realizations at the grid-point and regional scale would be nice to include. The other analysis that I would like to see is a comparison of the behavior of those time series/envelopes/red dots using the lowest scenario besides the highest (currently the only one presented). I think it will be interesting to evaluate if the differential behavior of true/emulated realizations remains qualitatively the same when comparing different scenarios. Right now, the scenario dimension is a bit downplayed by the choices of the validation metrics and I think it is too important an angle to be shortchanged. Otherwise, I really like the succinct metrics of validation used in the paper, it is never easy to synthesize these emulators' performance and the authors have done a nice job in that regard.

I have a few other comments which won't be as demanding. I will list those in the order they come up while reading the paper, even if some may be a little more substantial than others.

Page 2, lines 45-47: I know what is meant by this as we are all thinking about the same issues here, connecting our emulation work to the IAM community and their scenarios, but I do not think a random reader would understand/appreciate the problem. Maybe expand a bit, possibly describing a specific example (fires impeding the use of afforestation for carbon capture, for example). Some of this is mentioned on page 3, line 69 and following, maybe connect that discussion to this.

Page 2, line 50: TXx is not defined. Later on page 4, lines 93-94, annual maximum temperature is mentioned without a reference to TXx.

Page 3, line 54: maybe specify what periods were used for training? Historical and 21$^{st}$ century/SSPs I assume.

Page 5, line 118: the reference to Quilcaille et al., 2022 could call out MESMER-X explicitly.

Page 7, lines 172-173: I think the equations referenced should be (8) and (9) not (5) and (6). Also the quantities (Y in particular) in eq. 8 need to be defined.

Page 10, Figure 1 (but this also applies to the other similar figures): It is interesting to see light color cells for some of the ESMs for the reference stationary GEV (and implicitly Gaussian) distribution. **What are we to make of this**? I'm assuming this is a fit over both historical and scenario period, or is it just the historical used for training? Is the light color just a relatively better performance (but still bad) or is it somehow good? **Can the magnitude of the CRPS metric (in this case ~2.5 with little variation, in other figures much different) be interpreted**? In summary: I would like a bit more explanation of how to interpret the magnitude and performance of this baseline metric that is then used to say something about the relative performance of other choices of non-stationary distributions. Also, it is a bit unfortunate that light/dark colors have the opposite meaning in the first row and in the lower rows. Maybe calling this out in the caption could help the reader's not geeting confuse (but that reader is me, feel free to disregard this latter point).

Page 12, lines 264-265 I did not understand what is meant here: "The median of the ESM remains effectively at the center of the realizations by UKESM1-00-LL."

Page 15, lines 298-299: here is one spot where the interpretability of the magnitude of the CRPS would be of value.

Page 17, lines 321-324. I would argue that also the first-row panel (regional average) shows the same tendency. I think you mention it in the discussion, but would the model allow different choices (linear or quadratic) in different locations? If that is a possibility, I understand not giving it a try not to make things more complicated, but I think it could be mentioned here explicitly as a capability/powerful feature of the model…but maybe I'm wrong and it would be difficult to apply a mix of linear/non linear links?

Page 17, lines 335-336: I did not see this detailed point made at the beginning of the session. In this regard, could you also discuss if the use of a discrete distribution poses any challenge to the

probability transform, or if it all works seamlessly? Evidently it worked but it is not obvious to me how.

Page 17, lines 336-337: sentence needs correcting: "Using other distributions without distribution…"

Page 20-21: This behavior if soil moisture is really interesting! I'm impressed that just the lagged temperature is helping to account for this behavior. In that regard, is this lagged temperature produced by the emulator? Is it global temperature produced according to the scenario by a simple model? Need a bit more elaboration of how this is actually implemented. I'm also wondering if a more robust derivative measure (implicit in the use of the lagged T of course) could be a good auxiliary variable.

Page 26, lines 452-453: I could not understand this sentence: "While the range….assess variations".

---

## Referee Comment (RC2)

**Review of "Extending MESMER-X: A spatially resolved Earth system model emulator for fire weather and soil moisture by Quilcaille et al."**

**Summary**  This work introduces an extension of MESMER-X to produce regional emulation of fire weather and soil moisture indices. This is an important direction for emulation because these variables play a key role in the assessment of nature-based solutions to mitigate climate change.

The proposed emulators follow a shared construction. First, a local observation process $\mathcal{D}(\cdot|\alpha_{s,t,1}, \ldots, \alpha_{s,t,p})$ is prescribed to define the distribution of a variable of interest at time $t$ and spatial location $s$. Second, the parameters $\alpha_{s,t,1}, \ldots, \alpha_{s,t,p}$ of this observation process are estimated using (possibly non-linear) functions $f_{s,1}, \ldots, f_{s,p}$, which take as input global climate variables at time $t$. Internal variability is introduced by gaussianising the distribution $\mathcal{D}(\cdot|\alpha_{s,t,1}, \ldots, \alpha_{s,t,p})$, and introducing spatially-correlated innovations in this gaussianised space.

The model is evaluated for multiple indices of fire weather and soil moisture — which induce different choices of conditional distribution $\mathcal{D}$ and parametric functions $f_{s,p}$ — and demonstrates good performance, with limited quantile deviation.

**Strengths and Weaknesses**  Well written and presented paper, easy to follow. The model is formulated with great generality, which is a strength. Then, every single emulator is a particular case of the general formulation proposed. Extensive experiments are conducted on diverse variables which display different statistical properties and allow to fully appreciate the capacity of the model. Great effort is put into visualising the model performance.

In general, the emulators outperform the baseline and are capable to faithfully reproduce spatial patterns, even though the models local parametrisation are fairly simple, which is in my opinion a strength. The main weakness of the paper lies in the difficulty to assess the quality of models calibration on different future forcing scenarios. I would have appreciated to visualise emulation outputs on low and medium forcing scenarios.

I think the paper is very interesting and will be publishable after minor revision.

**Questions and comments**

- L41 : "For their part" $\rightarrow$ I'm assuming this is a literal translation from french "Pour leur part", might sound better to reformulate with for example "On the other hand"

- L115 : The functions $f_{s,1}, \ldots, f_{s,p}$ are time-independent, which introduces a time stationarity assumption for these mappings. From the results, this seems to be a robust assumption, but could you briefly comment on the grounds for this assumption?

- L122 : Could you comment on why would it be an appropriate model of internal variability to introduce innovations over the gaussianised variable? In particular, would be interested to hear about potential issues that could arise from the fact that the Normal distribution is not heavy tailed while the GEV distribution is (my intuition is that some tail events in a GEV density might not be properly represented by a Gaussian density, but that might be wrong).

- L180 : Probably the case already, but are score spatially averaged by accounting for decreasing cell size toward poles?

- Figure 1 (and other configuration selection plots): Could you add a label on the colorbars and make the fontsize larger for labels (the model description labels in particular are a bit difficult to read). It seems at first glance that CRPS values are quite high for $E_0$ (even lowest is around 2.2). That means that even for best CRPSS values, we still get a relatively high CRPS score for the emulator (this is also true for the other experiments). Could you comment on this?

- Figure 2, 7 : Could the caption be on the same page as the figure?
- Figure 3 (and other quantile deviation tables) : Is this computed jointly over multiple SSPs? I would assume so, but could benefit the manuscript to write this explicitly somewhere.
- L335 : "using other distributions without distribution"?
- L346 : "ESMs mostly provided the total soil moisture" → Suggestion : "a majority of ESMs only provide the total soil moisture"
- L359 : I think it would benefit the reader to provide intuition on why annual averages can be well represented by a normal distribution.
- L509-L514 : slightly redundant with the previous paragraph.

**Additional suggestions** *I do not expect these suggestions to be integrated in the revised manuscript.*

- It would be interesting to evaluate the fitness of the proposed conditional distributions with a statistical test (e.g. Kolmogorov-Smirnov test) or by evaluating the loglikelihood of ESM outputs under the proposed conditional distribution. I would expect the statistical test to fail because the fit would really need to be perfect, but the obtained p-value would nonetheless be a useful indicator.

- It would be useful to evaluate the soundness of the emulator with a calibration score (e.g. take the 95% confidence interval of your emulated distribution, and see what fraction of the observations fall within — it should be 95% of them). That should provide a concise and intuitive assessment of the emulators calibration, whilst the CRPS is a distance between cdfs which may be robust, but hard to develop practical intuition upon.

---

## Author Response (AR1)

**Response to Claudia Tebaldi, Referee 1, for the manuscript**

**Extending MESMER-X: A spatially resolved Earth system model emulator for fire weather and soil moisture**

Yann Quilcaille[1], Lukas Gudmundsson[1], Sonia I. Seneviratne[1]

[1]Institute for Atmospheric and Climate Science, Department of Environmental Systems Science, ETH Zurich, Zurich, Switzerland.

*Correspondence to*: Yann Quilcaille (yann.quilcaille@env.ethz.ch)

We would like to sincerely thank Claudia Tebaldi for the open, positive and constructive review. Her and the comments of the other Referee were completely integrated to the manuscript, which we believe has improve its quality.

We detail in this document the modifications brought to the manuscript. Referees' comments are shown in black. The authors' response is shown in green text. The text quoted from the manuscript is shown between quotation marks in italics. Numbers of lines correspond to the version including tracked changes.

Summary of modifications:
- Corrections in the main text following Referees' recommendations, as detailed in the responses to the Referees.
- Updated figures 1, 4, 8 & 11 (selection of configuration): reversed colormap on CRPS, labels on color bars and bigger font sizes
- New appendices:
    - 6.1: Application of the Probability Integral Transform to discrete distributions
    - 6.2: Representation of the interannual variability
    - 6.3: Interpretability of the CRPS
    - 6.4: Maps of negative log likelihood for the retained configurations
    - 6.5 to 6.8: Equivalent of figures 2, 5, 9, 12 but for SSP1-2.6 and SSP2-4.5

This paper details a modification/extension of the established MESMER-X approach to emulating spatially and temporally resolved impact-relevant variables. The targets of the emulations are historical and scenario outputs from CMIP6-era ESMs, focusing specifically on quantities relevant to wildfire risk, i.e., Fire Weather Index and soil moisture, and in particular measures of their tail behavior.

The paper is clearly written (maybe some occasional oddity due to non-native English usage, I know about that issue myself J), well organized and interesting. It shows the value and promise of the MESMER-X approach, highly relevant for integrating climate information and impact/mitigation analysis, and I think it will be publishable after some minor revision. A lot of what is presented is – as the title states – the extension of a methodological framework that has been established and peer reviewed already, so my comments do not question that part, but are mainly suggestions for further validation/clarification/expansion.

Thank you very much for expressing this opinion! Regarding the oddities, we will read the text another time for some corrections.

I start from perhaps the only significant request: I think it would be good to extend the validation in two directions: the first one is interannual variability, right now obfuscated by considering all realizations together and evaluating only the relative positions of target vs. emulated realizations in those plots showing many time series at once and their envelopes. Since interannual variability may be important in creating and making persistent some of these hazardous conditions, some simple metric that compares it between true and emulated realizations at the grid-point and regional scale would be nice to include. The other analysis that I would like to see is a comparison of the behavior of those time series/envelopes/red dots using the lowest scenario besides the highest (currently the only one presented). I think it will be interesting to evaluate if the differential behavior of true/emulated realizations remains qualitatively the same when comparing different scenarios. Right now, the scenario dimension is a bit downplayed by the choices of the validation metrics and I think it is too important an angle to be shortchanged. Otherwise, I really like the succinct metrics of validation used in the paper, it is never easy to synthesize these emulators' performance and the authors have done a nice job in that regard.

Thank you for these insightful comments. We agree that the two suggested directions would deserve more attention.
- Interannual variability indeed matters for the consequences of these hazards, but also to check how capable is MESMER-X to represent these aspects. The most direct approach we can think of is to represent temporal correlations. We have added an appendix with figures, representing the local correlations for one ESM and its emulated counterpart over three periods: 1851-1900; 2051-2100 of SSP1-2.6 and 2051-2100 of SSP5-8.5. We show that the inter-annual variability is correctly represented over preindustrial, low-warming and high-warming scenarios. This appendix is mentioned line 153.
- We acknowledge that we focused on the higher warming scenarios, to present the evolutions over larger domains of warming. The Anonymous Referee 2 has made similar suggestions. We have added the equivalent of figures 2, 5, 9 and 12 for SSP1-2.6 and SSP2-4.5, though in the appendix.

Page 2, lines 45-47: I know what is meant by this as we are all thinking about the same issues here, connecting our emulation work to the IAM community and their scenarios, but I do not think a random reader would understand/appreciate the problem. Maybe expand a bit, possibly describing a specific example (fires impeding the use of afforestation for carbon capture, for example). Some of this is mentioned on page 3, line 69 and following, maybe connect that discussion to this.

Good point, we have edited the text to include more details, as follows:
*"For instance, IAMs mitigate climate change by using bio-energies with carbon capture and storage (BECCS) and afforestation, yet these nature-based solutions would be impacted by droughts and fires (Fuss et al., 2014; Smith et al., 2016; Anderson and Peters, 2016). Thus, accurately replicating regional changes in climate extremes and water conditions of Earth System Models (ESMs) at a lower computational cost would help in exploring mitigation potentials and new emissions scenarios."*

Page 2, line 50: TXx is not defined. Later on page 4, lines 93-94, annual maximum temperature is mentioned without a reference to TXx.

The definition of TXx was a bit hidden just before the reference. We have edited the text to give it more visibility:

*"The MESMER emulator has been developed with this purpose, first for regional mean variables (Beusch et al., 2020; Beusch et al., 2022), and more recently also extended to the MESMER-X version representing TXx, the annual maximum temperatures (Quilcaille et al., 2022)."*

Page 3, line 54: maybe specify what periods were used for training? Historical and 21st century/SSPs I assume.
Yes, it makes sense. We have added the period:
*"Each one of these emulations account for the spatial and temporal correlations in TXx. MESMER-X was trained on each available ESM of the Climate Model Intercomparison Project Phase 6 (CMIP6) over 1850-2100 (Eyring et al., 2016; O'Neill et al., 2016)."*

Page 5, line 118: the reference to Quilcaille et al., 2022 could call out MESMER-X explicitly.
Agreed, we were not sure how to write that. We went for the following sentence:
"Similarly, if $\mathcal{D}$ is a GEV, equation (1) is equivalent to the formalism introduced in the article showcasing MESMER-X (Quilcaille et al., 2022)."

Page 7, lines 172-173: I think the equations referenced should be (8) and (9) not (5) and (6). Also the quantities (Y in particular) in eq. 8 need to be defined.
Thanks for pointing this out, it is corrected.

Page 10, Figure 1 (but this also applies to the other similar figures): It is interesting to see light color cells for some of the ESMs for the reference stationary GEV (and implicitly Gaussian) distribution. **What are we to make of this**? I'm assuming this is a fit over both historical and scenario period, or is it just the historical used for training? Is the light color just a relatively better performance (but still bad) or is it somehow good? **Can the magnitude of the CRPS metric (in this case ~2.5 with little variation, in other figures much different) be interpreted**? In summary: I would like a bit more explanation of how to interpret the magnitude and performance of this baseline metric that is then used to say something about the relative performance of other choices of non-stationary distributions. Also, it is a bit unfortunate that light/dark colors have the opposite meaning in the first row and in the lower rows. Maybe calling this out in the caption could help the reader's not geeting confuse (but that reader is me, feel free to disregard this latter point).
That's right, we didn't explain enough this part. Interpreting the CRPS is complicated by its lack of an upper value, while its minimum is zero. We brought more details when we introduce the metric (lines 179-182):
*"A high CRPS for this benchmark means that the differences between the cumulative distribution functions are too big, which implies that a stationary distribution does not correctly reproduce the statistical properties of the training sample, while a distribution reproducing perfectly the training sample would have a CRPS of zero (Hersbach, 2000), as illustrated with Figure A.1 in the Appendix."*

We have also edited the Figure 1, 4, 8 and 11 to reverse the colors of the CRPS, with an edit of the caption as follows:
***"Figure 1: Selection of the configuration for the seasonal average of the FWI (*FWIsa*).***
*For each ESM, the CRPS and CRPSS are averaged over space, time and scenarios. The darker is the colour of a cell, the better is the configuration at reproducing the distribution of the ESM. The upper row (white to black) corresponds to the CRPS of the configuration used as benchmark. A higher CRPS (lighter colour) indicates that the stationary distribution used as benchmark does not reproduce well the distribution of the ESM. The next rows (white to red)*

*correspond to the CRPSS of the tested configurations, relatively to the benchmark. A higher CRPSS (darker colour) indicates that the proposed configuration improves the reproduction of the distribution of the ESM."*

For information, the Referee 1 also asked for modifications on these figures, more precisely on the font sizes and titles of the color bars.

Page 12, lines 264-265 I did not understand what is meant here: "The median of the ESM remains effectively at the center of the realizations by UKESM1-00-LL."

Thank you for noting that. We corrected this sentence, hopefully now better:

*"Over 2014-2100, the realizations by UKESM1-0-LL remain mostly within the range of the emulations, except for the third row that corresponds to a grid point close to Manaus in Amazonia."*

Page 15, lines 298-299: here is one spot where the interpretability of the magnitude of the CRPS would be of value.

This is indeed a good spot to illustrate how to interpret this metric. Here is the text we added:

*"Because the higher is a CRPS, the worse is the distribution at representing the training sample, two results can be deduced. First, stationary GEV distributions are much better at reproducing $FWIsa$ than stationary Poisson distributions are at reproducing $FWIxd$. It may be because $FWIxd$ has stronger responses to climate change than $FWIsa$, meaning that stationary distributions, Poisson or GEV, cannot correctly reproduce these evolutions. It may also be because the shape of a Poisson distribution cannot reproduce as well the shape of the observed $FWIxd$, compared to a GEV for $FWIsa$."*

Page 17, lines 321-324. I would argue that also the first-row panel (regional average) shows the same tendency. I think you mention it in the discussion, but would the model allow different choices (linear or quadratic) in different locations? If that is a possibility, I understand not giving it a try not to make things more complicated, but I think it could be mentioned here explicitly as a capability/powerful feature of the model…but maybe I'm wrong and it would be difficult to apply a mix of linear/non linear links?

That's right, the top row shows such an effect, although to a lesser extent, because that's averaged over a region where not all grid points have this effect. We are adding that:

*"The same effect appears on the first row, although to a lesser extent."*

At the moment, MESMER-X assumes that all grid points share the same configuration (distribution & functions on parameters). Theoretically, MESMER-X would support having the configuration tailored to the grid point, and we already did some technical steps in this direction (looping over configurations & choice on CRPS or BIC), but not all of them (interpreting parametrizations that depend on the grid point). However, the full implementation of this feature isn't planned for this paper. Though, we are still unsure about the marginal gain in performances. We are adding that to the conclusion:

*"Making parametrizations dependent on the grid point would be a solution, but wasn't implemented for this article."*

Page 17, lines 335-336: I did not see this detailed point made at the beginning of the session. In this regard, could you also discuss if the use of a discrete distribution poses any challenge to the probability transform, or if it all works seamlessly? Evidently it worked but it is not obvious to me how.

We should have announced this point from the beginning of section 3.3, here is what we added lines 299-304:

*"Using this distribution implicitly assumes that the events are independent of each other, which is not exactly the case here. Assuming that a day matches the criteria for extreme fire weather (Quilcaille et al., 2023) for instance during the fire season, there are higher chances to have the next days also matching this criteria, compared to a period out of the fire season. Nevertheless, we choose this distribution because of its relative simplicity."*

You are right, a discrete distribution through a probability integral transform (PIT) isn't straight-forward. We added an appendix with a figure and some explanations, introduced in the manuscript, lines 135-136:

*"Equation (2) applies as well if $\mathcal{D}$ is a discrete distribution, as illustrated in Appendix 6.2."*

We do not copy here the full text of the appendix, that is a bit too long. In a nutshell, it works seamlessly because of two reasons. The first one is that a discrete distribution at a value X is representative of the interval [X-0.5; X+0.5[, there is an underestimation over [X-0.5; X[, but an overestimation over [X; X+0.5[, thus leading to partly compensating errors. The second one is that this process occurs in one way during training, then the other way round during emulation. We acknowledge that this is not a rigorous demonstration, but we are also planning to write down all the statistics behind to ensure that.

Page 17, lines 336-337: sentence needs correcting: "Using other distributions without distribution…"

Thanks for noting this error, here is the correction:

*"Using other distributions that would not assume independent events may improve these results but would require a higher degree of complexity."*

Page 20-21: This behavior if soil moisture is really interesting! I'm impressed that just the lagged temperature is helping to account for this behavior. In that regard, is this lagged temperature produced by the emulator? Is it global temperature produced according to the scenario by a simple model? Need a bit more elaboration of how this is actually implemented. I'm also wondering if a more robust derivative measure (implicit in the use of the lagged T of course) could be a good auxiliary variable.

We tried this effect to give a sense of the local trend in temperature. With $\Delta T$ being the change in global mean temperature, one may rewrite the terms as follows:

$$\lambda_{s,1}\Delta T_t + \lambda_{s,2}\Delta T_{t-1} = (\lambda_{s,1} + \lambda_{s,2})\Delta T_t + (-\lambda_{s,2})(\Delta T_t - \Delta T_{t-1})$$

Thus, the second term can be associated with the first derivative in time. We decided to write it down this way, because $\Delta T_t$ is our main driver, and introducing its derivative may confuse the readers. The lagged temperature is not produced by the emulator, it is simply the one at the former year that the ESM provides. For a scenario in 2015, we are using the corresponding historical in 2014. For the value in 1850 for its former year, we tried either using the average over 1850-1899 or the value of 1850 itself, but it does not make much difference, for it is just 1 point, with a preindustrial period long enough to account for this period. In this regard, we could start training in 1851 instead. To elaborate while not being too technical, we edited lines 395-397 as follows:

*"Here, as a first attempt to reproduce this effect, we will test in the configuration a lagged variable using the $\Delta T$ at the former year. This lagged variable is obtained by shifting the $\Delta T$ of the ESM by one year. From a modeling perspective, having both $\Delta T_t$ and $\Delta T_{t-1}$ is equivalent to having the value at year $t$ and its first derivative."*

We preferred here a backward difference operator for the first derivative, to give more weight to the past. we agree that other measures may be more appropriate. Without going into all the details, one could model the interannual change in the variable instead of the variable itself. Also, $\Delta T_{t-1}$, $\Delta T_{t-2}$,… and/or $\Delta T_{t-n}$ with a BIC criteria may help for different timescales. Extending this principle could be done using impulse response functions.

To investigate the proper modeling approach, we think that we have to identify adequate variables, and to try them on adequate scenarios, e.g. with overshoots. This is in our ToDo list, and we are convinced that such a work would benefit to all spatial emulators :)

Page 26, lines 452-453: I could not understand this sentence: "While the range….assess variations".
Thank you for noting that. We cleaned this sentence and the one after, leading to:
*"The spatial patterns of the ESM shown here on the top row, CNRM-CM6-1, are correctly reproduced by the emulations on the three following rows. The right column shows that the regional responses are correctly reproduced, with a majority of the ESM points being within the range of the emulations."*

**Response to the Anonymous Referee 2 for the manuscript**

**Extending MESMER-X: A spatially resolved Earth system model emulator for fire weather and soil moisture**

Yann Quilcaille[1], Lukas Gudmundsson[1], Sonia I. Seneviratne[1]

[1]Institute for Atmospheric and Climate Science, Department of Environmental Systems Science, ETH Zurich, Zurich, Switzerland.

*Correspondence to*: Yann Quilcaille (yann.quilcaille@env.ethz.ch)

We would like to sincerely thank the Anonymous Referee 2 for the positive and constructive review. Its and Claudia Tebaldi's comments were completely integrated to the manuscript, which we believe has improve its quality.

We detail in this document the modifications brought to the manuscript. Referees' comments are shown in black. The authors' response is shown in green text. The text quoted from the manuscript is shown between quotation marks in italics. Numbers of lines correspond to the version including tracked changes.

Summary of modifications:

- Corrections in the main text following Referees' recommendations, as detailed in the responses to the Referees.
- Updated figures 1, 4, 8 & 11 (selection of configuration): reversed colormap on CRPS, labels on color bars and bigger font sizes
- New appendices:
    - 6.1: Application of the Probability Integral Transform to discrete distributions
    - 6.2: Representation of the interannual variability
    - 6.3: Interpretability of the CRPS
    - 6.4: Maps of negative log likelihood for the retained configurations
    - 6.5 to 6.8: Equivalent of figures 2, 5, 9, 12 but for SSP1-2.6 and SSP2-4.5

**Summary** This work introduces an extension of MESMER-X to produce regional emulation of fire weather and soil moisture indices. This is an important direction for emulation because these variables play a key role in the assessment of nature-based solutions to mitigate climate change. The proposed emulators follow a shared construction. First, a local observation process $D( \cdot |\alpha_{s,t,1}, \ldots, \alpha_{s,t,p})$ is prescribed to define the distribution of a variable of interest at time $t$ and spatial location $s$. Second, the parameters $\alpha_{s,t,1}, \ldots, \alpha_{s,t,p}$ of this observation process are estimated using (possibly non-linear) functions $f_{s,1}, \ldots, f_{s,p}$, which take as input global climate variables at time $t$. Internal variability is introduced by gaussianising the distribution $D( \cdot |\alpha_{s,t,1}, \ldots, \alpha_{s,t,p})$, and introducing spatially-correlated innovations in this gaussianised space. The model is evaluated for multiple indices of fire weather and soil moisture — which induce different choices of conditional distribution $D$ and parametric functions $f_{s,p}$ — and demonstrates good performance, with limited quantile deviation.

Thank you very much for this review. We cannot help but notice the term that you introduce, "gaussianising". We appreciate it so much that we may borrow it from you, if you may! The first referee asked for details about this transformation, we added an appendix about it where we use this term, we hope that you don't mind.

**Strengths and Weaknesses** Well written and presented paper, easy to follow. The model is formulated with great generality, which is a strength. Then, every single emulator is a particular case of the general formulation proposed. Extensive experiments are conducted on diverse variables which display different statistical properties and allow to fully appreciate the capacity of the model. Great effort is put into visualising the model performance.

In general, the emulators outperform the baseline and are capable to faithfully reproduce spatial patterns, even though the models local parametrisation are fairly simple, which is in my opinion a strength. The main weakness of the paper lies in the difficulty to assess the quality of models calibration on different future forcing scenarios. I would have appreciated to visualise emulation outputs on low and medium forcing scenarios.

I think the paper is very interesting and will be publishable after minor revision.

We are grateful for your very positive comment. Regarding the outputs on other scenarios, we agree with you and Referee 1 that the scenario dimension may not have been clear enough in this manuscript. We used high warming scenarios to show the evolution over larger domains of warming, but showing their performances on low to mid warming scenarios do matter. We are adding the equivalent of the figures 2, 5, 9 and 12 for SSP1-2.6 and SSP2-4.5 to the Appendix.

- L41 : "For their part" → I'm assuming this is a literal translation from french "Pour leur part", might sound better to reformulate with for example "On the other hand"

Thank you for noting that, this is likely what happened… We corrected into:

*"Besides, changes in the water cycle are more challenging to represent than changes in temperature (Allan et al., 2020)."*

- L115 : The functions fs,1, . . . , fs,p are time-independent, which introduces a time stationarity assumption for these mappings. From the results, this seems to be a robust assumption, but could you briefly comment on the grounds for this assumption?

Thanks for this insightful comment. The initial reason why we use this assumption is because of pattern scaling. At a regional scale, we observe this linear scaling at a regional level, for instance between global mean temperature and regional temperatures. It was extended to the grid cell level for different applications, e.g. MESMER. What we did with MESMER-X is to extend this principle to conditional distributions. Though, even if this assumption is based on what we observe with the scaling in CMIP outputs, it does not explain the actual ground.

One way to look at the question would be to reason on its contraposition. If we assume that the mappings were not stationary in time, then it means that we could not isolate all dependencies of the parameters of the distribution being driven, here, by global mean temperature.

A potential reason may be that global mean temperature isn't enough. For instance, there may be some inertias in the water cycle, as seen with the soil moisture. Adding a lagged temperature helped there, but one may think of other forms of lagged variables or other variables like radiative forcing or ocean heat contents or fluxes. If we manage to rewrite the $f_{s,p}$ with these new global drivers, then we verify the initial assumption.

If this is still not enough, then it is not because of additional global drivers or dependencies on the global pathway. It means that at a prescribed global climate, the parameters of the distributions are not fixed. Because the global climate pathway would also be prescribed, it is not due to changes in regimes / bifurcation / hysteresis effects. As far as we can think of it, the

only remaining effect would be local drivers, such as changes in land use or different "mix in radiative forcings", for instance more greenhouse gases but also more from cooling aerosols. Such effects would lead to similar global climate pathways, but without the same local effects. As a summary, this assumption allows to account for a large range of global effects, but its major issue would be changes in local drivers that would compensate at the global scale. Though, fixing that isn't straight-forward, as the modified model may need local inputs as well, thus hindering its use for coupling with simple climate models. A step in this direction has been made in (Nath et al., 2022).

We think that this discussion may be of interest to some readers, so we added the following paragraph lines 121-128. We are afraid that it may not have been as "briefly" as you asked for.

*"Equation (1) offers a large flexibility in terms of modeling. Using variables such as global mean surface temperature, radiative forcing or ocean heat content facilitates the modeling of interplays in the Earth system. Using lagged variables such as the global mean temperature at $\Delta T_{t-n}$ or accumulated warming over the past $n$ years would also help in representing more advanced dynamics such as inertias in the water cycle. Such a capacity is of particular interest for overshoot scenarios. Yet, equation (1) has also its limits: any changes in local climate drivers (e.g. land-use, combination of individual radiative forcings) that would compensate at a global scale would not be accounted for. Such effects may still be modeled (Nath et al., 2022), but are not integrated in this framework."*

- L122 : Could you comment on why would it be an appropriate model of internal variability to introduce innovations over the gaussianised variable? In particular, would be interested to hear about potential issues that could arise from the fact that the Normal distribution is not heavy tailed while the GEV distribution is (my intuition is that some tail events in a GEV density might not be properly represented by a Gaussian density, but that might be wrong).

Thanks for this excellent comment. The first reason is that representing internal variability on the non-gaussianised variable would be much harder. Detrending is feasible, but then we would also need to deal with changes in spread, eventually in the shape of the distribution, that is not trivial. That is why we made this modeling choice.

Then, regarding the choice of this specific model of internal variability over the gaussianised variable, we think that it would work because this probability integral transform is a projection of the cumulative distributive function. It implies that the tails would be correctly reproduced. By construction, the distribution of gaussianised events follow a Normal distribution, which ensures that generating events using this model of internal variability, based on Normal distributions, would normally give a proper Normal distribution, then a proper reconstruction of the tails. More precisely, the tail events at 99% or higher in the GEV would occur in the 99% or higher region of the Normal distribution as well. When generating realizations, about 1% would be in this region of the Normal, thus only 1% in the GEV.

The Referee 1 had also a question regarding this transformation, though on discrete distributions. This is why I added an appendix, with a figure for some visual material.

- L180 : Probably the case already, but are score spatially averaged by accounting for decreasing cell size toward poles?

Thanks again for this comment. Yes, we are accounting for the cell size. The CRPS is calculated in each grid point and at each time step of each scenario, for each its member. Then it is averaged over ensemble members, so that scenarios run more than others would not have a higher weight in the final performances. Then, we average over time, accounting for the length of the historical or scenario. Finally, we average over space, accounting for the size of the grid cells, as you said, so that the cells at the poles don't artificially bias the performances. The CRPSS is calculated in each grid point, time step, each member, then averaged the same way.

- Figure 1 (and other configuration selection plots): Could you add a label on the colorbars and make the fontsize larger for labels (the model description labels in particular are a bit difficult to read). It seems at first glance that CRPS values are quite high for E0 (even lowest is around 2.2). That means that even for best CRPSS values, we still get a relatively high CRPS score for the emulator (this is also true for the other experiments). Could you comment on this?
We acknowledge that these labels were too small, and that titles for the color bars would help. We have edited the figures 1, 4, 8 and 11 to integrate these requests. For information, the Referee 1 also asked to revert the colors on the CRPS, for a better interpretability.
Regarding the interpretability of the CRPS, our understanding is that it is due to the large natural variability in the obtained values. For instance, you point to $FWIsa$, with a lowest average CRPS of 2.2. The $FWIsa$ has an approximate range of 10 for an approximate value of 20. We highlight that these approximations are very rough, only for the sake of explanation. Given this spread, a realization by the ESM would likely have a high CRPS value. The average over time steps would still lead to a high CRPS. To give some statistical ground to this explanation, the equation 8.55 p.353, of (Wilks, 2011a) for the CRPS a Normal distribution of location $\mu$ and scale $\sigma$ and an observation X can be written:

$$CRPS = \sigma \left( X(2\,\mathcal{F}_{\mathcal{N}}(X, \mu, \sigma) - 1) + 2f_{\mathcal{N}}(X, \mu, \sigma) - {1}/{\sqrt{\pi}} \right)$$

With $f$ and $\mathcal{F}$ the respective probability and cumulative distribution functions of the Normal distribution. Thus, assuming variables with different scales but with points at the same quantiles of the distribution, the variables with the higher scales would have the higher CRPS. The Referee also asked for information on the interpretability of the CRPS, so we are adding the appendix 6.1.

- Figure 2, 7 : Could the caption be on the same page as the figure?
This is a good point, we will make sure that it happens during the editing of this manuscript before publication.

- Figure 3 (and other quantile deviation tables) : Is this computed jointly over multiple SSPs? I would assume so, but could benefit the manuscript to write this explicitly somewhere.
Yes, they are calculated together over the available scenarios, and then averaged. We are adding this information in the caption of the Figure 3:
*"The deviation is calculated on all available scenarios."*

- L335 : "using other distributions without distribution"?
Thanks for noting this error, here is the correction:

*"Using other distributions that would not assume independent events may improve these results but would require a higher degree of complexity."*

- L346 : "ESMs mostly provided the total soil moisture" → Suggestion : "a majority of ESMs only provide the total soil moisture"
Changed accordingly to your suggestion.

- L359 : I think it would benefit the reader to provide intuition on why annual averages can be well represented by a normal distribution.
We have simply added a mention to the theorem that was implicit:

"As an annual average, *SM* may be represented by a normal distribution according to the central limit theorem."

- L509-L514 : slightly redundant with the previous paragraph.
We decided to keep this paragraph, because even if it is slightly redundant, it summarizes the former paragraph, which is what some readers may want to read.

It would be interesting to evaluate the fitness of the proposed conditional distributions with a statistical test (e.g. Kolmogorov-Smirnov test) or by evaluating the loglikelihood of ESM outputs under the proposed conditional distribution. I would expect the statistical test to fail because the fit would really need to be perfect, but the obtained p-value would nonetheless be a useful indicator.
That is an excellent point. We have appended a section with figures for the negative log-likelihood, averaged over the size of the training sample. We chose the NLL instead of the Kolmogorov-Smirnov, because distributions are trained by minimizing the NLL. We are adding a reference in the appendix the first time that we identify a configuration, likes 259-260:

*"We point out that the local performances for this configuration are shown in the Appendix 6.7, along with those of the other variables emulated."*

It would be useful to evaluate the soundness of the emulator with a calibration score (e.g. take the 95% confidence interval of your emulated distribution, and see what fraction of the observations fall within — it should be 95% of them). That should provide a concise and intuitive assessment of the emulators calibration, whilst the CRPS is a distance between cdfs which may be robust, but hard to develop practical intuition upon.
We agree that showing a concise, intuitive but accurate as well assessment of the emulator is the golden target. This is complicated by the number of dimensions (space, time & scenarios, quantiles, configurations, ESMs). This is why we had created the figures 3, 6, 10 and 13, for the deviations of the quantiles. We think that the readers may use these figures to evaluate the performances of the emulator.

**Response to Vivek Arora, editor for the manuscript**

**Extending MESMER-X: A spatially resolved Earth system model emulator for fire weather and soil moisture**

Yann Quilcaille[1], Lukas Gudmundsson[1], Sonia I. Seneviratne[1]

[1]Institute for Atmospheric and Climate Science, Department of Environmental Systems Science, ETH Zurich, Zurich, Switzerland.

*Correspondence to*: Yann Quilcaille (yann.quilcaille@env.ethz.ch)

Thank you very much for reading this manuscript and your comments. We consider that taking into account your suggestions will indeed improve the readability for a broader audience, and not only for those familiar with climate emulators.

We detail in this document the modifications brought to the manuscript. Your comments are shown in black. The authors' response is shown in green text. The text quoted from the manuscript is shown between quotation marks in italics. Numbers of lines correspond to the version including tracked changes.

Line 103, what is timesteps t ? Is this the annual time step of the data from the comprehensive climate models?

This is a good point, the method has been validated for annual time steps but not others. We edited as follows:

"Given data of a climate variable $X_{s,t}$ in grid points $s$ and annual time steps $t$,"

(edited later for your comment on line 160).

Line 111 and 113. Vt is defined as "covariants" on line 112 and "changes in global climate" on line 113. I am able to follow line 113 but not sure what covariants mean in this context.

Thanks for pointing out that we did not define the term "covariant". We edited as follows:

"Explicitly, the $p$ parameters $\alpha_{s,t,p}$ of $\mathcal{D}$ at grid points $s$ are functions $f_{s,p}$ of a matrix of global variables $V_t$. The columns of the matrix $V_t$ contain covariants, explanatory variables such as global mean temperature anomalies, while the rows of $V_t$ correspond to time steps."

Line 123. It seems "innovations" is emulator terminology. I am unable to follow what "spatially correlated innovations" mean.

We acknowledge that "spatially correlated innovations" is a statistical term, that many readers may not be familiar with. We propose the following explanation, hopefully helping the readers:

"To integrate these effects, we follow the approach of (Beusch et al., 2020), that parametrizes internal climate variability using the statistical process described in (Humphrey and Gudmundsson, 2019). Temporal correlations are represented by an auto-regressive process (equation 3), while the spatial correlations are reproduced with spatially correlated innovations (equations 4 to 6)."

Line 132. "... we obtain the equivalent of normalized residuals ...". Residuals from what?

Thank you for pointing this out. These residuals referred to the residual variability line 124, where the auto-regressive process is usually applied of a detrended variable, thus its residuals. Because the Probability Integral Transform generalizes its use as described in the lines before, we think that the term "residuals" is not necessary here We rewrite the sentence as follows:

"We then employ the probability integral transform, obtaining a normalized variable $\Phi_{s,t}$, where $\Phi_{s,t}$ has no trend and follows a standard normal distribution"

And the line 135 is now corrected as well:

"The normalized variable $\Phi_{s,t}$ are then characterized using an autoregressive process with spatially dependent innovations (Beusch et al., 2020)."

Line 160. "... require timeseries of anomalies in global climate". Do you mean anomalies from historical climate? If yes, what time period.

Thank you for this comment. We acknowledge that the time period for the training data was defined neither in this section Method, nor on the sections Data. This is why we clarify this point by editing the beginning of Section 2.2:

"We introduce the climate variable $X_{s,t}$ in grid points $s$ and annual time steps $t$. Typically, $X_{s,t}$ is deduced from CMIP6 historical and SSP scenarios, covering 1850-2100 and the whole Earth."

The line 160 is then edited as follows:

"The emulations of climate extremes for a scenario, typically over 1850-2100, require time series of anomalies in global climate $V_t$ over the period of the scenario, so that equation (1) generates the distributions at each grid point and each time step."

Equation (8). Unless, I missed it, it seems Y_s,t,n is not defined.

You are right, it was a former notation of the emulations, now written $\tilde{X}_{s,t,n}$. Thanks for noting.

Similar to reviewers may I please suggest to describe in words what does CRPS actually quantifies.

We acknowledge that the CRPS wouldn't be familiar to all readers. On top of our additions following the reviewers' recommendations, we are adding a simple interpretation. We hope that the following sentence would give a sense to the readers of what the CRPS quantifies:

"To assess and compare the performances, we use the ensemble Continuous Rank Probability Score ($CRPS$), a generalization of mean absolute errors for probabilistic forecasts. The $CRPS$ measures differences in the cumulative distribution functions of the emulations $\tilde{X}_{s,t,n}$ and of the training data $X_{s,t}$ (Hersbach, 2000; Wilks, 2011b)."

Equation (10). Have the terms, mu, sigma, and xi (skewness?) been defined?

Thank you for reminding us that the notations for the coefficients have not been introduced. We add an appropriate description of these notations before the figures on the CRPS (1, 4, 8 and 11), where they appear for the first time.

Please consider y-axis labels for Figure 2 (column c) and other similar figures.

Thank you for your comment. We have been trying to include a label in the figures 2, 5, 9 and 12 (and the new figures A.11 to A.18 asked by the reviewers for low & mid warming). However, it has an overall negative impact on the readability of the figure. This is mostly due to the "annual minimum of

the monthly average soil moisture" written four times in a single figure. The alternative would have been writing its term $SMmm$, yet we would need to change it also for the columns (a) and (b). We prefer to have the explicit name of the variable over the term, so that readers going straight for the figures would still understand what is represented.

This is why we are sorry not to include the y-axis labels for this column in these figures.

It seems, the ESMs x regions are not defined/introduced.

Thank you for pointing this out, we didn't introduce this term. We have edited the sentence that introduces this figure and these regions:

"Figure 3 provide more details on the deviation of quantiles of MESMER-X for each ESM and land region (Iturbide et al., 2020), thereafter called ESMs x regions."

Line 314. Please change Figure 4 to Figure 5.

We thank you for your careful reading. This is indeed a typo, that is now corrected.

Lines 316 and 317. This sentence about the HadGEM3-GC31-MM model refers to which figure.

Thanks for noting that. We have added the following precisions in the sentence:

"For instance, in 2014 (Figure 5a), HadGEM3-GC31-MM returns higher $FWIxd$ to the south of Sahel, but lower in South America. In 2100 (Figure 5b) in the centre of Africa and in South-East Asia, we see differences in these patterns, though the emulations always relatively similar."

Section 4. Please introduce units of soil moisture. My recollection is that soil moisture units are kg/m2. Yet in the figures the units of soil moisture are kg/m3. Please check. It also seems that you have included both liquid and frozen soil moisture in your analysis (hence the maximum on your colour scales goes to 5000). Can you please make this clear in the manuscript?

You are right, this is here the total soil moisture, hence water in all phases, summed over all soil layers. We decided the total soil moisture instead of only the liquid component or only in the soil layer, because less CMIP6 runs provided the other variables, restraining the number of ESMs or ensemble members for training. We have detailed this choice in Section 4.1:

"We base the annual indicators for soil moisture on the total soil moisture content (CMIP6 variable $mrso$). Ideally, soil moisture in the root zone would be more relevant to investigate droughts. Thus, soil moisture in soil layer (CMIP6 variable $mrsos$ or $mrsol$) would have been more adapted (Qiao et al., 2022). Similarly, the total soil moisture content includes all water phases, thus frozen soil moisture as well. We deem that the total soil moisture content remains relevant for droughts, in regions without high frozen soil moisture, that is to say not higher latitudes or mountainous regions like the Himalaya. Nevertheless, a majority of ESMs only provide the total soil moisture content, thus choosing this variable ensures that the capacity of the emulator can be evaluated on more models and ensemble members."

Regarding the unit, this is indeed kg/m2, not m3. Sorry for such a mistake, it was indeed some very heavy water. We have corrected the units in figures 7, 9, 12 and the new ones A15 to A18. For information, we tried non-linear color scales, it did improve the contrast, but at the cost of the readability of the values.

Section 4.2. In the context of lagged response of soil moisture to deltaT, I am wondering if precipitation could be factor here. Recall that higher warming also implies more precipitation so it is possible that higher precipitation in RCP 8.5 scenario is helping soil moisture not decrease as fast despite higher deltaT, compared to the other scenarios. If yes, please consider noting this.

Thanks for this comment. We agree that precipitations would factor here, the higher precipitations would be an effect of the imbalance in the atmospheric energy budget (Shine et al., 2015). However, the rest of water cycle would also contribute. While there would be higher precipitations, there would also be higher evapotranspiration. (Allan et al., 2020) shows that there is a slow feedback response both in land evapo(transpi)ration and in global precipitations (first, decrease, then increase), due to thermal imbalances. The time scales of such responses would be of several years. We add this note in Section 4.2:

"To a broader extent, this effect is related to the response of the whole water cycle, with rapid adjustments and slow feedback responses, both in precipitations and evapotranspiration(Allan et al., 2020)."

In Figure 1, the darkest colours are associated with the highest values of CRPSS for the HadGEM3 model. Can you please clarify what does this implies? Does it mean that emulators fit this model the best?

Thank you for pointing this out. In few words, it only means that there was more room for improvements for this ESM, that there was a stronger climate signal to represent instead of a stationary distributions. With more details, the HadGEM3 models have a CRPSS higher than most of the other CMIP6 ESMs, but it had a CRPS also higher than most. The CRPSS being defined relatively to the stationary distribution used in the CRPS, the high CRPS indicates that a stationary distribution were worse at representing HadGEM3 models than the other ESMs, suggesting a relatively stronger climate signal. However, the CRPSS higher than for other ESMs indicate a stronger margin of improvement on this ESM. Yet, its final performance are not necessarily better than for the other ESMs. For instance, on HadGEM3 models, with roughly a CRPS of 2.55 and a CRPSS of 0.2, the CRPS of the emulators would be 2.55 * (1 - 0.2) = 2.04. Another ESM like CMCC-CM2-SR5 starts with roughly a CRPS of 2.25 and a CRPSS of 0.10, thus the CRPS of the emulators would be 2.25 * (1 - 0.10) = 2.025.

This discussion is summarized with the following sentence after Figure 1:

[revised manuscript text omitted]

Wilks, D. S.: Statistical methods in the atmospheric sciences, Academic press2011a.

Wilks, D. S.: Statistical methods in the atmospheric sciences, 2011b.

---

## Author Response (AR2)

**Extending MESMER-X: A spatially resolved Earth system model emulator for fire weather and soil moisture**

Yann Quilcaille[1], Lukas Gudmundsson[1], Sonia I. Seneviratne[1]

[1]Institute for Atmospheric and Climate Science, Department of Environmental Systems Science, ETH Zurich, Zurich, Switzerland.

*Correspondence to*: Yann Quilcaille (yann.quilcaille@env.ethz.ch)

Dear Vivek Arora,

Thank you for your positive review of this manuscript and your attentive reading. We have integrated your comments in this new version.
I am grateful to you, such a valuable editing process represents an important experience to any researcher, and even more to early career scientists.

Line 19: Please reword. Calculating how much do quantiles deviate doesn't result in good performance but rather allows to evaluate performance.
Thanks for pointing out to this shortcut. We have rephrased this sentence as follows:
"For each of the four variables considered, we evaluate the performances of the emulations by calculating how much do their quantiles deviate from those of the ESMs."

Line 45: IAMs don't mitigate climate change, they simulate mitigate climate change using bio-energies.
This is indeed another shortcut, that we have corrected as follows:
"For instance, IAMs simulate the mitigation of climate change by using bio-energies with carbon capture and storage (BECCS) and afforestation."

Line 48: simulated by
This is indeed an useful precision.
"Thus, accurately replicating regional changes in climate extremes and water conditions simulated by Earth System Models (ESMs) at a lower computational cost would help in exploring mitigation potentials and new emissions scenarios."

Line 100: with
Line 104: represent; for; and AT annual time steps
All of these suggestions were implemented.

Line 123-124: Please consider rewording, perhaps - the representation of primary interactions within the Earth system.
This sentence was rephrased more adequately:

"Using variables such as global mean surface temperature, radiative forcing or ocean heat content facilitates the representation of the most relevant processes within the Earth system."

Line 136: This is the first time you have used the term "spatially correlated innovations". Can you please define it here?
We have rewritten several sentences in this paragraph, for a better explanation of what this statistical process entails:
"To integrate these effects, we follow the approach of (Beusch et al., 2020), that parametrizes internal climate variability using the spatially autoregressive (SAR) noise model described in (Cressie and Wikle, 2011; Humphrey and Gudmundsson, 2019). The SAR model reproduces the temporal and spatial autocorrelation structure of the training data, using two components. Temporal correlations are represented by an auto-regressive process (equation 3). Spatial correlations are reproduced with spatially correlated innovations, randomly generated from a multivariate Gaussian with zero mean and covariance matrix derived from the training sample (equations 4 to 6)."

Line 147: Please consider rewording. Perhaps something like ... equation (2) works equally well if D is a discrete distribution
We have followed your suggestion.

Line 148-149: Did you mean "spatially correlated innovations"?
Thanks for noting this issue, we have corrected.

Line 151: It seems r has not been defined yet. It's defined later on next page in equation (6).
We acknowledge that r was defined later. We delayed this definition to privilege the continuity of the introduced elements, yet it was unclear with r. We follow your suggestion and add this sentence:
"Here, $r$ designs the ratio of geographical distance between points and a localization radius, and the next paragraphs explaining how $\Sigma_\nu(r)$ is obtained from the empirical covariance matrix."

Line 236: Shouldn't this be FWIsa?
Thank you very much for pointing out this typo! We have corrected that.

Line 244: I think, these haven't been introduced before. Although obvious you do not to tell these are mean and std dev.
These parameters were indeed not explicitly introduced before, only implicitly in this sentence. We have rephrased this sentence for an explicit definition, and added the link to the definition of the first equation related to the conditional distributions.
"For a normal distribution, the parameters $\alpha$ introduced in equation (1) are the location and scale, written respectively $\mu$ and $\sigma$ in Figure 1, corresponding to the mean and standard deviation of the distribution. For a GEV distribution, the parameters $\alpha$ are the location, scape and shape, written respectively $\mu$, $\sigma$ and $\xi$ in Figure 1."

Line 326-327: Doesn't Poisson distribution has only one parameter - lambda, and it's mean is in fact lambda?

The basic Poisson distribution has only this lambda as a parameter. However, having this sole parameter constrains too much the distribution. This is why the additional parameter mu allows for more freedom, through a shift of the distribution. We have edited this sentence to reflect this aspect.

"Here, the parameters $\alpha$ introduced in equation (1) are the rate $\lambda$ and a shift $\mu$. The training of the distribution gains in freedom using this shift of the distribution by $\mu$, with its mean becoming $\mu + \lambda$, while the variance remains $\lambda$."

Another important modification of the manuscript was the modification of the Competing Interests:

"Yann Quilcaille, Lukas Gudmundsson and Sonia I. Seneviratne declare that they do not have any competing interests. Though, for the sake of transparency, we notify that Sonia I. Seneviratne is a member of the editorial board of Earth System Dynamics. However, we point out that it had no impact on the reviewing and editing processes of this manuscript, for which the handling editor is Vivek Arora."

Beusch, L., Gudmundsson, L., and Seneviratne, S. I.: Emulating Earth system model temperatures with MESMER: from global mean temperature trajectories to grid-point-level realizations on land, Earth Syst. Dynam., 11, 139-159, 10.5194/esd-11-139-2020, 2020.
Cressie, N. and Wikle, C. K.: Statistics for spatio-temporal data, John Wiley & Sons, Hoboken, New Jersey, USA, 624 pp.2011.
Humphrey, V. and Gudmundsson, L.: GRACE-REC: a reconstruction of climate-driven water storage changes over the last century, Earth Syst. Sci. Data, 11, 1153-1170, 10.5194/essd-11-1153-2019, 2019.

---

## Author Response (AR3)

**Extending MESMER-X: A spatially resolved Earth system model emulator for fire weather and soil moisture**

Yann Quilcaille[1], Lukas Gudmundsson[1], Sonia I. Seneviratne[1]

[1]Institute for Atmospheric and Climate Science, Department of Environmental Systems Science, ETH Zurich, Zurich, Switzerland.

*Correspondence to*: Yann Quilcaille (yann.quilcaille@env.ethz.ch)

Dear Vivek Arora,

Thank you for your comments, we do think that they have contributed in improving this manuscript. The points 1 and 2 have been implemented as suggested, while we slightly adapted the suggestions made in the point 3.
We remain at your disposal, would there be anything else to correct.

1.
Note the suggested changes in UPPERCASE letters below.

Here, $r$ REPRESENTS the ratio of geographical distance between points and a localization radius, and the next FEW paragraphs EXPLAIN how $\Sigma \nu(r)$ is obtained from the empirical covariance matrix.
The suggested changes have been implemented, thank you!

2.
Note the suggested changes in UPPERCASE letters below. I changed "scape" to "scale".

For a normal distribution, the parameters $\alpha$ introduced in equation (1) are the location and scale, written respectively $\mu$ and $\sigma$ in Figure 1, corresponding to the mean and standard deviation of the distribution. For a GEV distribution, the parameters $\alpha$ are the location, SCALE, and shape, written respectively $\mu$, $\sigma$ and $\xi$ in Figure 1.
Thanks for pointing out this typo.

3.
In the following sentence

Here, the parameters $\alpha$ introduced in equation (1) are the rate $\lambda$ and a shift $\mu$. The training of the distribution gains in freedom using this shift of the distribution by $\mu$, with its mean becoming $\mu + \lambda$, while the variance remains $\lambda$

you have defined $\lambda$ as both a rate and as variance. Can you please revise this

sentence? In the usual Poisson distribution the minimum value is zero, and the mean and standard deviation are $\lambda$. When you shift it by $\mu$ does the minimum value become $\mu$?

Here's one suggestion. Not sure if this satisfies what you intended to describe. Please check.

Here, the parameters $\alpha$ introduced in equation (1) for the Poisson distribution are $\lambda$ and a shift $\mu$ in mean . This modified form of the Poisson distribution has the same variance ($\lambda$) as the original Poisson distribution but the mean is $\lambda+\mu$ (as opposed to $\lambda$), and the minimum value is $\mu$ (as opposed to zero).

I acknowledge that this sentence wasn't clear enough. I have adapted your suggestion as follows:
Here, the parameters $\alpha$ introduced in equation (1) are the rate $\lambda$ and a shift $\mu$ in the distribution. This modified form of the Poisson distribution has the same variance ($\lambda$) as the original Poisson distribution, but the mean is $\mu + \lambda$ instead of $\lambda$. This shift in the Poisson distribution allows for more flexibility in the training.